# Charge polarity-dependent ion-insertion asymmetry during electrochemical doping of an ambipolar π-conjugated polymer

Jibin J. Samuel[1], Ashutosh Garudapalli [1], Chandrasekhar Gangadharappa[1], Smruti Rekha Mahapatra[1], Satish Patil [1] & Naga Phani B. Aetukuri [1]✉

Electrochemical doping is central to a host of important applications such as bio-sensing, neuromorphic computing and charge storage. However, the mechanisms that enable electrochemical dopability and the various parameters that control doping efficiencies are poorly understood. Here, employing complementary electrochemical and spectroelectrochemical measurements, we report a charge-polarity dependent ion insertion asymmetry in a diketopyrrolopyrrole-based ambipolar π-conjugated polymer. We argue that electrostatic interactions are insufficient to fully account for the observed charge-specific ion insertion into the polymer matrix. Using polymer side-chain dependent electrochemical doping studies, we show that electron density donating and accepting tendencies of polymer side-chains sufficiently describe the observed charge-polarity dependent electrochemical doping. Our observations are akin to the solvation of dopant ions by polymer side-chains. We propose that Gutmann donor/acceptor number framework qualifies the 'solvent-like' properties of polymer side-chains and provides a rational basis for designing π-conjugated polymers with favorable mixed ionic electronic transport properties.

π-conjugated polymeric (π-CP) semiconductors are a class of technologically important materials with wide-ranging applications[1]. The rich structure-property space of π-CPs, accessible through organic synthesis techniques, allows material properties to be tailored to specific applications. For example, π-CPs find applications in organic thin-film transistors[2], organic light-emitting diodes[3], and organic photovoltaics[4]. Favorable electronic and/or optoelectronic properties of π-CPs are key to their utility in these applications. Electrochemical modulation of π-CPs' electronic and/or optoelectronic properties via electrochemical doping could further enable a host of novel applications such as in bioelectronics[5], neuromorphic computing[6], electrochromics[7], energy storage[8], and electroactuation[9].

During electrochemical doping, redox of a π-CP and an associated charge transfer is concomitant with the insertion and de-insertion of ions into the bulk of the π-CP from an adjoining electrolyte. Ion insertion and carrier injection into the π-CP occur simultaneously

during electrochemical doping. The net result is a coupling of ionic fluxes with the electronic conductivity of the material. The resultant mixed ionic electronic conduction and an associated modification of electronic/optoelectronic properties of the π-CP is central to realizing functionalities of relevance for the aforementioned applications.

However, fundamental mechanisms that lead to successful electrochemical doping of a π-CP are still unclear. For example, most known π-CPs afford electrochemical-doping induced hole transport (p-type electrochemical doping). Surprisingly, very few π-CPs are known to afford n-type electrochemical doping. Remarkably, n-type electrochemical doping was not reported in π-CPs that have high n-type carrier mobilities. Further, polymer design parameters such as polymer chain length[10], side-chain functional groups[11–13], passive/active swelling[14] of the polymer and polymer backbone rigidity[13] seem to affect electrochemical doping. In addition, electrochemical parameters such as dopant ion size and polarity and electrolyte solvent

[1]Solid State and Structural Chemistry Unit, Indian Institute of Science, Bengaluru 560012 Karnataka, India. ✉e-mail: phani@iisc.ac.in

could also affect electrochemical doping. Whether these parameters predominantly affect ion insertion and ionic conduction or carrier mobilities and electronic conduction or both is unclear.

Conventional theories of doping in inorganic semiconductors or molecular doping in polymeric semiconductors need not be sufficient for understanding electrochemical doping of π-CPs. This is because the nature and strength of electrostatic interactions between dopant ions or between ions and induced electronic charge carriers are not only a function of the π-CP's dielectric constant but are also influenced by the 'dynamically configurable' local polymer environment around these charges[15,16]. Additionally, ion concentration or electrochemical dopant density is continuously changing during electrochemical doping with dynamic doping being an inherent feature of electrochemical devices. It is therefore challenging to develop mechanistic insights and identify descriptors that affect electrochemical doping.

In this work, we study the tendency for ion insertion during electrochemical doping for several spherically symmetric ions. Ion size-dependent ion insertion and resultant conductivity changes are probed using an organic electrochemical transistor (OECT) geometry. OECT platform enables the simultaneous probing of electrochemically-driven ion insertion and the resultant change in the channel's electronic conductance with easily accessible gate voltages of -1 V in magnitude. Further, it is rather straightforward to vary electrochemical parameters such as ion size, polarity, electrolyte solvent and the channel material. We advantageously use this simple experimental design to selectively change either the ion size, polarity or the electrochemical channel material to glean mechanistic insights into ion insertion.

For most of the studies reported in this work, we chose 2DPP-OD-TEG as the choice channel material. 2DPP-OD-TEG is a donor-acceptor (D-A) π-conjugated redox polymer based on diketopyrrolopyrrole (DPP) acceptor and thiophene (T) donor moieties. Figure 1a shows the chemical structure of 2DPP-OD-TEG wherein the lactam nitrogen atoms of T-DPP-T units are functionalized alternately with triethylene glycol (TEG) and 2-octyl dodecyl (OD) side-chains. Further, 2DPP-OD-TEG is redox active for both anion and cation insertion thereby allowing the study of charge polarity-dependent ion insertion during electrochemical doping.

Based on electrochemical doping studies, we found that anion insertion during electrochemical doping of 2DPP-OD-TEG is strongly correlated with ion size while cation insertion is size-independent. The charge polarity-dependent ion insertion tendency is rather surprising and cannot be explained using any simple electrostatic models. Our findings suggest that the polymer favoring cation or anion insertion correlates with the electron-density donating or accepting ability of the side-chain functional groups, respectively. We propose that the interactions between the dopant ions and the polymer matrix may be modeled analogously to ion-solvent interactions in electrolyte solutions and that the Gutmann donor and acceptor numbers could be useful guiding metrics for π-CP design.

## Results

### N- and p-type OECT characterization in aqueous 0.1 M NaCl electrolyte

OECT measurements were performed in the configuration shown in Fig. 1b (see methods section for more details). This is similar to a 3-electrode electrochemical cell and enables accurate measurement of gate voltages without interference from potential shifts associated with electrochemical currents flowing across the gate and source electrodes.

First, we discuss the n-type operation of 2DPP-OD-TEG OECTs in aqueous 0.1 M NaCl electrolyte. OECT transfer characteristics, i.e., the plot of $I_D$ as a function of applied gate voltage $V_{G,REF}$, is shown in Fig. 1c. As $V_{G,REF}$ is increased beyond the n-type threshold voltage ($V_{T,n}$), the channel undergoes an electrochemical reduction reaction which corresponds to the insertion of Na$^+$ ions from the electrolyte and a concomitant injection of electrons from the source-drain contacts. This results in electrochemical n-doping of the channel with a clear turn-ON with channel current increasing for $V_{G,REF} > V_{T,n}$.

Next, the OECTs were characterized for p-type OECT operation in aqueous 0.1 M NaCl electrolyte. $V_{G,REF}$ is scanned to negative voltages to induce channel oxidation while a bias voltage $V_{DS} = -0.4$ V was applied across the channel. However, even at gate voltages as high as $-1.2$ V, the OECTs did not show a distinct p-type turn ON: $I_D$ was within a few tens of nA (Fig. 1d). We found that gold contacts were not

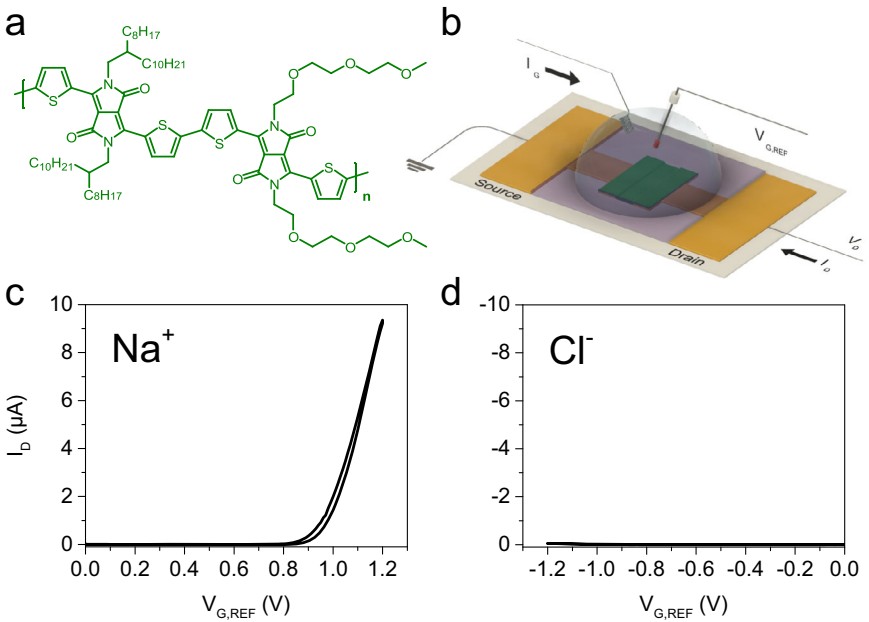

**Fig. 1 | 2DPP-OD-TEG-based n- and p-type OECT operation. a** Chemical structure of 2DPP-OD-TEG polymer. **b** Measurement configuration used for OECT measurements. Typical n-type (**c**) and p-type (**d**) OECT transfer characteristics of 2DPP-OD-TEG OECTs measured in 0.1 M aqueous NaCl. Transfer characteristics were measured by scanning the gate voltage at a scan rate of 20 mV/s from 0 to 1.2 V with the drain-source bias voltage ($V_{DS}$) set to 0.4 V for n-type and from 0 to $-1.2$ V at $V_{DS} = -0.4$ V for p-type electrochemical doping.

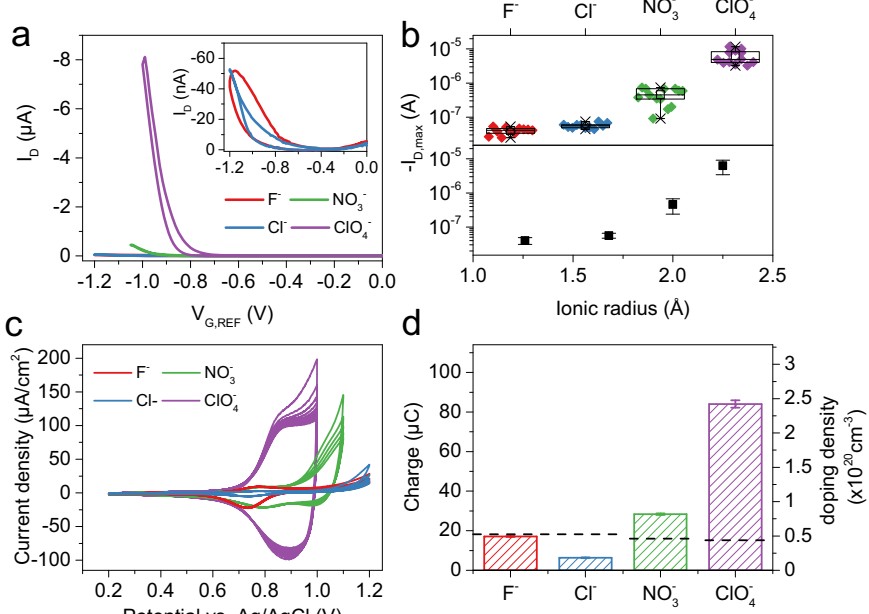

**Fig. 2 | Anion size dependence for p-type electrochemical doping of 2DPP-OD-TEG. a** Typical p-type OECT transfer characteristics of 2DPP-OD-TEG OECTs measured in 0.1 M aqueous solutions of sodium salts with different anions as mentioned in the plot legend. **b** Box plot showing $I_{D,max}$ measured for identical OECTs for insertion of different anions along with plot of mean $I_{D,max}$ as a function of the ionic radius of the respective inserted anion. Error bars represent standard deviation. **c** Five cycles of CV at a scan rate of 100 mV/s in electrolytes containing each of the four anions from 0.2 V to $V_{CV,max}$ corresponding to $V_{G,max}$ used for OECT measurements in Fig. 2a. **d** Mean charge under the reductive wave of the CVs and corresponding volumetric doping density. Dashed black lines indicate the charge that would correspond to charging of typical electrostatic double layer ($C_{EDL} = 40 \, \mu F/cm^2$) at respective $V_{CV,max}$. Error bars represent standard deviation. For the box plots, center line is median; box limits are 25th and 75th percentiles; whiskers are outliers within 25th and 75th percentile + 1.5x interquartile range; '□' represents mean value; '×' represents maximum and minimum values.

electrochemically stable at such high oxidative potentials. Therefore, platinum contacts were used for all OECTs in this work.

The lack of p-type OECT operation is surprising considering the following factors together. (1) 2DPP-OD-TEG shows ambipolar conduction in organic field effect transistor (OFET) geometry[17] which implies sufficient mobilities of electrons and holes, (2) the frontier energy levels of the polymer (Supplementary Information (SI) section 1 and Supplementary Table 1) are comparable to the electrochemical stability window of the aqueous electrolyte which implies that both the n- and p-doped states must be stable despite contact with the electrolyte[18], and (3) Cl⁻ ions have a lower hydration energy (−344 kJ/mol) compared to Na⁺ (−383 kJ/mol)[19] which implies it should have been easier for Cl⁻ ions to insert into the polymer matrix to dope it p-type[20].

### Ion size dependence for p-type OECT operation

To rationalize the lack of p-type electrochemical doping of 2DPP-OD-TEG with an aqueous NaCl electrolyte, we consider a two-step mechanism that leads to conductivity modulation of the polymer during electrochemical doping. The first step is dopant-ion insertion with an associated polymer redox and carrier injection into the polymer channel. Second, electrochemically induced charge carriers must be sufficiently mobile to move in response to an external electric field. Whether one or both of these steps limit the conductivity of the channel is unclear. To disentangle the contributions from these two steps, electrochemical doping experiments were performed with different anions.

Sodium salts with F⁻, Cl⁻, NO₃⁻, and ClO₄⁻ anions were used to prepare 0.1 M aqueous electrolyte solutions for p-type OECT measurements so as to cover a wide range of ion sizes with ionic radii ranging from 1.26 to 2.25 Å[21]. Note that the charge density distribution is nearly spherically symmetric for all the ions considered here. P-type OECT transfer characteristics were measured using the configuration

shown in Fig. 1b. Gate voltage, $V_{G,REF}$, was swept at a rate of 20 mV/s from 0 V to a negative gate voltage $V_{G,REF,max}$ upto which stable OECT operation was observed for each of the anions ($V_{G,REF,max} = -1.2$ V for F⁻ and Cl⁻; −1.05 V for NO₃⁻ and −1 V for ClO₄⁻). Drain-source bias was set to $V_{DS} = -0.4$ V for these measurements.

The maximum drain current, $I_{D,max}$ was measured to be of the order of tens of nA for electrochemical-doping with smaller anions: F⁻ and Cl⁻. $I_{D,max}$ values of ~500 nA were measured for OECTs gated through the NO₃⁻ containing electrolyte (see Fig. 2a). However, significantly higher mean $I_{D,max}$ values of ~6 μA were observed for electrochemical doping with the ClO₄⁻ anion. 10–15 identical OECTs were measured with each of the four electrolytes to obtain a statistical distribution of various p-type OECT parameters. Box plots of $I_{D,max}$ for each of the anions for all devices (top panel of Fig. 2b) show that the measured values are highly reproducible across several devices (also see Supplementary Figs. 3, 4).

Bottom panel of Fig. 2b shows the mean value and standard deviation of $I_{D,max}$ plotted as a function of ion size. $I_{D,max}$ shows a strong correlation with ion size and increases with increasing anion size. $V_{T,p}$, shows a general shift toward less negative voltages with increasing dopant anion size (see Supplementary Fig. 4c). Flagg et al.[20] have observed a similar ion size-dependent p-type operation for poly 3-hexyl thiophene (P3HT) based OECTs. They showed that OECTs gated through electrolytes containing larger anions such as ClO₄⁻ and PF₆⁻ turned ON, but those gated through electrolytes with small anions such as F⁻ and Cl⁻ showed low to no channel conductivity.

P-type OECTs showing low $I_{D,max}$ with the smaller anions could be either due to the absence of ion insertion or due to the low mobility of carriers in the presence of smaller anions[22]. To identify the microscopic mechanisms that lead to the observed ion size dependence, we performed complementary experiments to quantify ion insertion.

## Cyclic voltammetry and electrochemical impedance spectroscopy measurements

Cyclic voltammograms (CVs) of 2DPP-OD-TEG films in each of the 4 anion-containing electrolytes are shown in Fig. 2c. CVs were obtained at a scan rate of 100 mV/s between 0.2 V and a maximum potential, $V_{CV,max}$ of 1.2 V, 1.2 V, 1.05 V, and 1 V vs. Ag/AgCl for F$^-$, Cl$^-$, NO$_3^-$, and ClO$_4^-$ respectively (also see Supplementary Fig. 7a–d). $V_{CV,max}$ corresponds to $V_{G,max}$ used for the p-type OECT measurements with different anions (see Fig. 2a). During the forward scan from 0 to $V_{CV,max}$, the polymer is expected to undergo oxidation with a simultaneous insertion of anions. During the backward scan, the polymer film returns to its neutral state along with the expulsion of anions. For CVs performed with ClO$_4^-$ containing electrolytes, an oxidative wave was observed on the forward scan with a corresponding reduction peak on the reverse scan, with a peak current density of ~−90 µA/cm$^2$. CVs with the smaller ions show smaller oxidative and reductive currents on the forward and backward scans respectively. In the case of NO$_3^-$ containing electrolytes, an oxidative wave and two reductive peaks of ~ −20 µA/cm$^2$ and −15 µA/cm$^2$ were observed around 1.0 V and 0.8 V respectively suggesting weak redox activity of the polymer corresponding to the insertion/de-insertion of NO$_3^-$ anions. For the smaller F$^-$ and Cl$^-$ anions, small redox peaks are observed centered around 0.7 V.

A quantitative estimate of the volumetric density of inserted ions can be obtained from the charge transferred during the CV scans. The current in the oxidative wave comprises of currents arising out of reversible processes such as charging of the electrostatic double layer (EDL) at the polymer/electrolyte interface and ion insertion into/oxidation of the bulk of the polymer film and irreversible processes such as parasitic side reactions (e.g., electrolyte decomposition – O$_2$ evolution). During the reductive scan, only the charge corresponding to reversible processes is recovered and this provides a lower bound for the number of ions that were originally inserted into the polymer film. The magnitude and fraction of the irreversible charge is quantified as a function of the CV upper vertex potential for all the ions (see Supplementary Fig. 7e, f). Parasitic reactions can reduce and de-dope the polymer film but form only a small fraction of the total charge and hence are not expected to significantly affect the ion-insertion and doping processes. Therefore, the central conclusions of ion-size dependence are unaffected by parasitic reactions, if any (also see SI section 3).

Figure 2d shows the mean value of the charge under the reductive wave of the CV averaged over 5 cycles. The corresponding doping density is shown on the right axis (assuming 100% doping efficiency). The dashed lines represent the equivalent charge that could be typically stored in a EDL at $V_{CV,max}$ for each of the electrolytes considering the upper limit of typical EDL capacitance of 40 µF/cm$^2$. In the case of F$^-$ and Cl$^-$, the total charge under the reductive wave was calculated to be 17 and 6.3 µC. This is not distinguishable from the charge that could be stored in the EDL (18 µC at 1.2 V, electrode area = 0.38 cm$^2$) at the polymer/electrolyte interface.

However, for NO$_3^-$ and ClO$_4^-$, the total charge recovered on the reductive scan is significantly higher than that corresponding to EDL charging. This confirms that volumetric ion insertion does indeed take place in the case of NO$_3^-$ and ClO$_4^-$. The charge under the reductive wave for ClO$_4^-$ ions at 1 V is 84 µC which corresponds to a volumetric doping density of $2.4 \times 10^{20}$/cm$^3$. This value is higher than that for NO$_3^-$ (28 µC or $8.0 \times 10^{19}$ /cm$^3$) at 1.05 V. The doping density follows the trend observed for I$_{D,max}$ as a function of the dopant anions (Fig. 2b). However, we note that the relation is not linear. For example, the corresponding OECT I$_{D,max}$ in the case of ClO$_4^-$ is 12 times higher than for NO$_3^-$ change while the estimated densities of inserted ions are ~3 times higher. This is not surprising given the known super linear relation between doping density and conductivity in organic semiconductors[23].

The influence of ion-size on the ion insertion/de-insertion dynamics is also quantified by electrochemical impedance spectroscopy (EIS). Bode magnitude and phase plots obtained for each anion at different gate voltages are shown in Supplementary Fig. 8. The effective volumetric capacitance ($C^*_{eff}$) is estimated from EIS measurements using the formula, $C^*_{eff} = -1/2\pi f Z_{im}$ where $Z_{im}$ is the imaginary component of the impedance response. $C^*_{eff}$ calculated at 0.1 Hz for p-type electrochemical doping is plotted as a function of applied potential from 0.5 to 1 V in Supplementary Fig. 9 (see experimental methods for details). The highest oxidative potentials were limited to 1 V because of large parasitic reactions observed at higher potentials. For electrochemical doping with ClO$_4^-$, $C^*_{eff}$ increases strongly with increasing oxidative potential from 2 F/cm$^3$ at 0.5 V to a maximum of 117 F/cm$^3$ at 0.9 V. By contrast, the change in $C^*_{eff}$ is relatively small for the other anions with increasing potentials. $C^*_{eff}$ measured at 1 V (vs. Ag/AgCl) corresponding to insertion of F$^-$, Cl$^-$, NO$_3^-$, and ClO$_4^-$ are 3.9, 19.8, 6.1, and 101 F/cm$^3$ respectively. $C^*_{eff}$ for Cl$^-$ being relatively higher than for F$^-$ may be an artefact arising from using the formula, $C^*_{eff} = -1/2\pi f Z_{im}$ in the presence of parasitic currents (see Supplementary Fig. 7e, f). Thus, OECT, CV, and EIS measurements are consistent with conductivity modulation for p-type electrochemical doping of 2DPP-OD-TEG being limited by ion insertion.

## Ion size dependence for cation insertion

Next, we investigated the effect of ion size on cation insertion during n-type electrochemical doping of 2DPP-OD-TEG. Aqueous 0.1 M electrolyte solutions of chloride salts with Li$^+$, Na$^+$, K$^+$, and Cs$^+$ as cations were used to measure n-type OECT characteristics of the polymer. All ions have a spherically symmetric charge distribution with ionic radii ranging from 0.73 to 1.81 Å[24], thereby covering a similar range as the size of the anions.

Figure 3a shows the n-type transfer characteristics of a typical OECT measured in each electrolyte. The gate voltage, $V_{G,REF}$, was swept from 0 V to a maximum gate voltage $V_{G,REF,max}$ upto which stable OECT operation was observed at a rate of 20 mV/s for each of the cations ($V_{G,REF,max}$ = 1.2 V for Li$^+$, Na$^+$; 1.1 V for K$^+$, Cs$^+$). Drain-source bias was fixed at $V_{DS}$ = 0.4 V for all measurements. 10-15 identical OECTs were measured for each of the electrolytes. As shown in box plots (top panel of Fig. 3b) of $I_{D,max}$ versus ion size, the measured drain currents are highly reproducible across several devices (also see Supplementary Figs. 10 and 11). Remarkably, the ion size dependent changes in the maximum drain current, $I_{D,max}$, for cation insertion are negligible (see bottom panel Fig. 3b) when compared to the 3-orders of increase in drain current seen during anion insertion. $V_{T,n}$ shows a general shift towards less positive voltages with increasing dopant cation size (Supplementary Fig. 11c) but the variation is smaller than that seen with the anions (Supplementary Fig. 4c).

Cyclic voltammetry experiments followed by EIS experiments were performed to quantify cation insertion into the polymer. All CVs showed a reductive wave on the forward scan and an oxidative peak on the backward scan indicating reversible insertion/de-insertion for all four cations (Fig. 3c). The total charge under the oxidative wave of the CV represents the portion of the charge deposited during the reductive scan which can be reversibly extracted and sets a lower bound on the estimated doping density. The mean charge under the oxidative wave of the CV are 47, 79, 72, and 42 µC corresponding to an estimated lower bound of $1.3 \times 10^{20}$, $2.3 \times 10^{20}$, $2.1 \times 10^{20}$, and $1.2 \times 10^{20}$ /cm$^3$ for the n-type doping density for Li$^+$, Na$^+$, K$^+$, and Cs$^+$ cations respectively at −1 V vs. Ag/AgCl. Further, $C^*_{eff}$ calculated from EIS measurements increases with increasingly negative potentials reaching values of 139, 135, 132, and 103 F/cm$^3$, at a potential of −1 V vs. Ag/AgCl for Li$^+$, Na$^+$, K$^+$, and Cs$^+$ insertion respectively (Fig. 3d and Supplementary Fig. 12). The changes in doping density and $C^*_{eff}$ as a function of cation size are insignificant compared to the changes observed for anion insertion. Clearly, CV and EIS experiments suggest that cation insertion is ion size

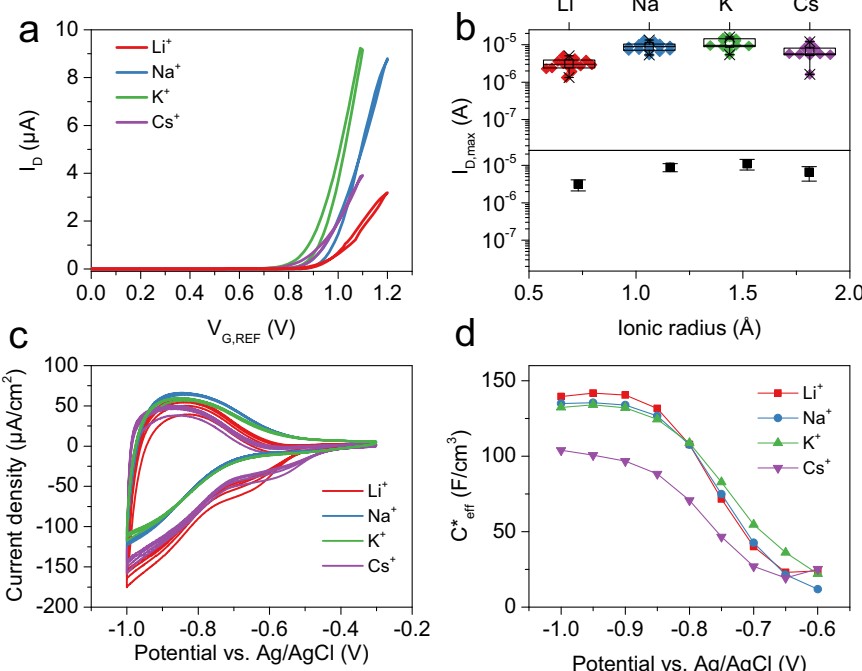

**Fig. 3 | Cation size dependence for n-type electrochemical doping of 2DPP-OD-TEG. a** Typical n-type OECT transfer characteristics of 2DPP-OD-TEG OECTs measured in 0.1 M aqueous solutions of chloride salts with different cations as mentioned in the plot legend. **b** Box plot showing $I_{D,max}$ measured for identical OECTs for insertion of different cations along with plot of mean $I_{D,max}$ as a function of the ionic radius of the respective inserted cation. Error bars represent standard deviation. **c** Five cycles of CV at a scan rate of 100 mV/s in electrolytes containing each of the four cations. **d** Volumetric capacitance, $C_{eff}^*$ estimated from EIS for different cations. For the box plots, center line is median; box limits are 25th and 75th percentiles; whiskers are outliers within 25th and 75th percentile + 1.5x interquartile range; '□' represents mean value; '×' represents maximum and minimum values.

independent, which is in stark contrast to the strong correlation between ion size and ion insertion tendency observed for anions. As the ionic radii is correlated with hydration energy and hardness of the ion, our observations can also be interpreted in terms of hydration energy or the Pearson hardness. We further discuss this in SI section 7.

All the cations used are monoatomic cations with comparable polarizabilities. To confirm that n-type electrochemical doping in 2DPP-OD-TEG is indeed not dependent on ion-size, OECT, CV, EIS experiments were performed with a large polyatomic cation, tetra butyl ammonium (TBA$^+$) with an ionic radius of 4.13 Å[25]. OECTs gated through 0.1 M TBACl electrolyte solution showed n-type operation with an $I_{D,max} = 462 \pm 165$ nA (at $V_{G,REF} = 0.9$ V), and an OECT turn-on voltage of $V_{T,n} = 0.69 \pm 0.03$ V(also see SI section 8 and Supplementary Figs. 14a–c). The mean charge under the oxidative wave of CVs is 146 μC corresponding to a doping density of ~4.2 × 10^20 /cm³ at −1 V And, $C_{eff}^*$ estimated from EIS measurements performed at −1 V is 136 F/cm³ (Supplementary Fig. 14d). The charge density and $C_{eff}^*$ values are of the same order as for other cations studied in this work suggesting that size independent electrochemical doping is observed even for TBA$^+$, a polyatomic cation.

### Changes in optical absorption during p- and n-type electrochemical doping

π-CPs have strong optical absorption at wavelengths corresponding to band-gap transitions. During electrochemical doping, the induced carriers change the relative absorbance at different wavelengths. Therefore, to further confirm the charge-polarity dependent ion insertion asymmetry in 2DPP-OD-TEG, we probed the changes in spectroscopic signatures of the polymer over the ultraviolet-visible-near infrared (UV-vis-NIR) wavelengths (300–1800 nm) during electrochemical doping with each of the ions used in this study.

Figure 4 shows the evolution of the relative absorption spectra, $\Delta Abs$ (relative to the spectrum obtained at 0 V) for both anions and

cations (also see Supplementary Fig. 16). As shown in Fig. 4a, quenching of the main absorbance peak and a corresponding increase in polaron absorption band ($\lambda > 1000$ nm) is most prominent for ClO$_4^-$ anion. The general features of the evolution of the spectra were similar to that reported earlier for electrochemical doping of DPP-based polymers[26,27]. The spectra obtained for F$^-$ and Cl$^-$ anions showed hardly any changes with increasing oxidative potential. In the case of NO$_3^-$, a weak signature of doping is observed suggesting a small extent of NO$_3^-$ insertion. In line with the CV and EIS observations, the spectroelectrochemistry experiments show that extent of redox of the polymer and corresponding ion insertion is the largest for ClO$_4^-$, negligibly small for F$^-$ and Cl$^-$, and intermediate for NO$_3^-$ anions. This further corroborates the anion size dependent ion insertion inferred from CV, EIS, and OECT experiments.

By contrast, quenching of the main absorbance peak and emergence of the polaron band were observed for all the 4 monoatomic cations (Fig. 4b) as well as for TBA$^+$ (Supplementary Fig. 14e). Based on spectroelectrochemistry experiments, we conclude that both ion insertion and the subsequent induction of mobile charge carriers are size-independent for n-type electrochemical doping and cation insertion. On the other hand, anion size strongly influences anion insertion and the formation of mobile charge carriers during p-type electrochemical doping. The OECT, CV, EIS, and spectroelectrochemistry experiments show a dopant charge-polarity dependent asymmetry in dopability of 2DPP-OD-TEG. Our observations cannot be trivially explained with any known point-charge based electrostatic theories. We propose an electrolyte-theory like framework to rationalize the observed charge polarity-dependent differences in ion insertion.

### Ion insertion energetics during electrochemical doping of π-CPs

We consider n-type electrochemical doping as a two-step electrochemical reaction comprising of polymer reduction followed by cation

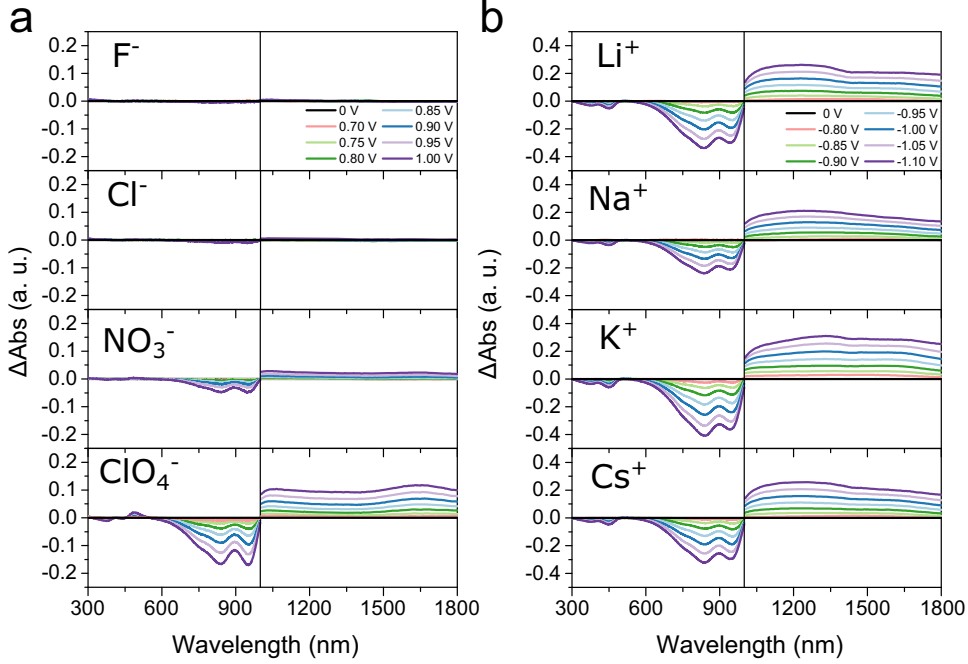

**Fig. 4 | Evolution of UV-vis-NIR absorption spectra of 2DPP-OD-TEG films during electrochemical doping as a function of the applied redox potential. a** UV-vis-NIR spectra collected during p-type electrochemical doping from electrolytes containing different anions at oxidative potentials ranging from 0 V to 1 V in steps of 50 mV. **b** UV-vis-NIR spectra during n-type electrochemical doping from electrolytes containing different cations at reductive potentials ranging from 0 V to −1.1 V in steps of 50 mV.

insertion:

$$P + e^- \rightarrow P^-; \Delta G = \Delta G^0_{red} \qquad (1)$$

$$A^+_{sol} \rightarrow A^+_{Poly}; \Delta G = \Delta G_{A^+,sol \rightarrow Poly} \qquad (2)$$

The standard free energy change for a 1e⁻ reduction reaction is given by $\Delta G^0_{red} = -FE^0_{red}$, where $F$ is the Faraday constant and $E^0_{red}$ is standard reduction potential of the polymer. The insertion of cations may be viewed as the transfer of solvated ions ($A^+_{sol}$) into the polymer matrix ($A^+_{Poly}$) with the associated free energy of transfer, $\Delta G_{A^+,sol \rightarrow Poly}$. Approximating solvents as structureless dielectric media, the free energy of ion transfer between two solvents can be estimated using the Born model of ion solvation. For the specific case of transfer of a cation, $A^+$ of charge $Z$ and ionic radius $r_{A^+}$ from an electrolyte solution with a dielectric constant, $\varepsilon_{sol}$, to the polymer with a dielectric constant $\varepsilon_{Poly}$, the Born free energy of transfer, $\Delta G^{Born}_{A^+,sol \rightarrow Poly}$ is given by[28]:

$$\Delta G^{Born}_{A^+,sol \rightarrow Poly} = -\frac{(Ze)^2}{8\pi\varepsilon_0 r_{A^+}} \cdot \left(\frac{1}{\varepsilon_{sol}} - \frac{1}{\varepsilon_{Poly}}\right) \qquad (3)$$

Noting that $\varepsilon_{r,water} = 78.5 >> \varepsilon_{r,polymer} \cong 2–5$, the value of $\Delta G^{Born}_{A^+,sol \rightarrow Poly}$ will be >0. In this case, ion insertion process incurs an energy penalty which can be represented by the insertion overpotential, $\eta_{ins} = -\Delta G^{Born}_{A^+,sol \rightarrow Poly}/F$ which decreases with increasing ion size. In line with this prediction of the Born model, we observed an inverse correlation between the ion size and the magnitude of p-OECT and n-OECT threshold voltages, $V_{T,p}$ and $V_{T,n}$. The $Z^2$ term in the Born free energy expression (Eq. 3) implies that the ion insertion energetics must be charge-polarity agnostic. Surprisingly, as discussed earlier, we observed that anion insertion is ion size dependent while cation insertion is not.

Noting that the ions considered are spherically symmetric and the solvent for all experiments considered here is water, we hypothesize that the observed deviation from the Born model originates from the structure of the polymer and charge-specific coordinating interactions between the polymer and ions. For example, the Born model does not differentiate between a four-fold and a sixfold coordinated ion in a solvent environment. Further, non-linear effects like dielectric saturation or polymer-ion interactions such as the coordination of the ions by the polymer matrix are not accounted for in the Born model[28].

Here, we propose that the Gutmann donor and acceptor numbers of the side-chain functional groups could be useful descriptors that account for specific coordinating interactions between the ions and the polymer matrix. As we discuss next, within this proposed framework, side-chains are akin to solvating moieties for inserted ions and enable ion stabilization in the polymer matrix.

## Gutmann donor and acceptor numbers of side-chain groups to qualify ion insertion energetics

Gutmann donor number (DN) and acceptor number (AN) are widely used solvent parameters to predict the ability of solvents to solvate ions[29]. Solvation could be described as a Lewis acid-Lewis base interaction[30]. The DN correlates with the Lewis basicity or electron density donating ability (EDD) of solvent molecules and qualifies the ability of the solvent to stabilize cations in solution. The AN correlates with the Lewis acidity or electron density accepting (EDA) ability of solvent molecules and qualifies the ability of the solvent to stabilize anions in solution.

The free energy change for cation transfer, $\Delta G^{transfer}_{A^+,sol \rightarrow Poly}$, from the electrolyte into the polymer during electrochemical doping can be expressed in terms of the difference in the DN values of the polymer ($DN_{Poly}$) and the solvent ($DN_{sol}$)[31] according to Eq. 4 (see SI section 10 and ref. 32 for details). The parameter $a_{A^+}$ is ion-specific and reflects the sensitivity of $\Delta G^{transfer}_{A^+,sol \rightarrow Poly}$ to the difference in DN values for the ion $A^+$. In general, the magnitude of $a_{A^+}$ correlates inversely with cation size, and the transfer free energy is more sensitive to the donor number difference for smaller cations. Ions could retain a part of their solvation shell during ion insertion. In this case, the ions can be treated as 'dressed' ions with a higher effective radius and a lower effective

Lewis acidity/basicity, depending on ionic charge. In case of cations, such co-insertion of solvent molecules can be accounted by readjusting the $a_{A^+}$ parameter, such that magnitude of $a_{A^+}$ increases with decreasing effective radius and increasing Lewis acidity.

$$\triangle G_{A^+,sol\rightarrow Poly}^{transfer} = a_{A^+} \cdot (DN_{Poly} - DN_{sol}) \qquad (4)$$

Similarly, for anions, the free energy change for anion transfer correlates with the difference in AN values of the two solvents[33]. Therefore, $\triangle G_{B^-,sol\rightarrow Poly}^{transfer}$ for an anion $B^-$ from the electrolyte with acceptor number $AN_{sol}$ to the polymer matrix with acceptor number $AN_{Poly}$ is given analogously to Eq. 4 by:

$$\triangle G_{B^-,sol\rightarrow Poly}^{transfer} = b_{B^-} \cdot (AN_{Poly} - AN_{sol}) \qquad (5)$$

Notably, the magnitude of $b_{B^-}$ is expected to be inversely correlated with anion size implying that $\triangle G_{B^-,sol\rightarrow Poly}^{transfer}$ is more sensitive to the difference in AN values for the insertion of smaller anions.

To rationalize the experimentally observed asymmetry in ion insertion, we consider the microstructure of 2DPP-OD-TEG and other π-CPs, which have a lamellar microstructure, composed of alternating layers consisting of π-π stacked polymer backbones and side-chain rich regions. The side-chains generally consist of long and flexible molecular chains which can flex to accommodate dopant ions. The ions inserted during electrochemical doping are likely to be accommodated in these side-chain rich regions[34]. Therefore, we believe that side-chains play a vital role in the coordination and stabilization of ions in the polymer matrix[35].

We first consider the TEG side-chains of the 2DPP-OD-TEG polymer. These are comparable to the series of ethylene glycol dimethyl ether molecules – glyme, diglyme, triglyme, and tetraglyme with DN = 18.6, 19.2, 14.0, 16.6 kcal mol$^{-1}$ respectively[36]. Thus the DN for TEG side-chains, $DN_{TEG} \approx 14$ - 19.2 kcal mol$^{-1}$, is comparable to the DN of water ($DN_{sol} \approx 18$ kcal mol$^{-1}$)[37]. Since the alkyl (OD) side-chains in 2DPP-OD-TEG are poorly coordinating, the effective DN of the polymer matrix could be expected to be largely determined by DN of TEG side-chains i.e. $DN_{Poly} \cong DN_{TEG}$. Since the term $\left(DN_{Poly} - DN_{sol}\right) \cong 0$ in Eq. 4, the free energy change for transfer of cations is expected to be very small and independent of ion size, despite $a_{A^+}$ having a strong inverse correlation with cation size[31]. (see SI section 10, Supplementary Table 2). This is consistent with cation size-independent ion insertion observed in the electrochemical doping of 2DPP-OD-TEG.

On the other hand, water is a much stronger EDA (AN$_{water}$ = 54.8) and stabilizes small anions much better than the TEG side-chains (AN for TEGDME = 10.5). The contribution of OD side-chains to anion stabilization could be effectively ignored due to low acceptor numbers (AN = 0 for hexane) of long-chain alkane solvents[38]. Since the difference in AN$_{poly}$ and AN$_{sol}$ is large, a large energy penalty for insertion of small anions is expected based on Eq. 5. However, as suggested earlier, $b_{B^-}$ is inversely correlated with anion size and therefore $\triangle G_{B^-,sol\rightarrow Poly}^{transfer}$ decreases with increasing anion size[39]. This, we suggest, gives rise to the observed strong ion size dependence for anion insertion (Fig. 2).

## OECT measurements – ion size dependence in a TEG free polymer

To further establish the veracity of the proposed DN/AN framework for ion insertion in polymers, we considered electrochemical doping of a DPP-based polymer with all alkyl-based side-chains. For these experiments, we replaced TEG side-chains in 2DPP-OD-TEG with hexyl side-chains and refer to the modified polymer as 2DPP-OD-HEX (see Fig. 5a). Since the side-chain regions are composed entirely of alkyl groups which have a low DN ($DN_{Poly} \sim 0$), the term $DN_{Poly} - DN_{sol}$ (see Eq. 4) is a large negative number. Consequently, unlike for cation insertion in 2DPP-OD-TEG, the change in free energy for cation transfer to 2DPP-OD-HEX is expected to be sensitive to the parameter $a_{A^+}$, which

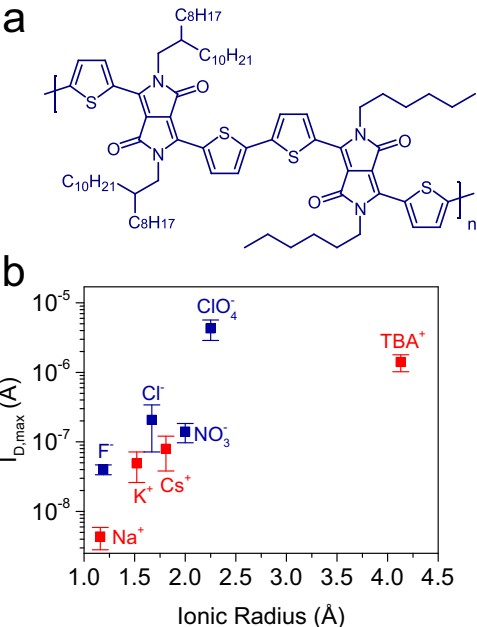

**Fig. 5 | N- and p-type OECTs using 2DPP-OD-HEX, a polymer with all alkyl side-chains. a** Chemical structure of 2DPP-OD-HEX polymer repeat unit. **b** Mean drain current for p-OECT and n-OECT operation as a function of ionic radii of inserted ions. Error bars represent standard deviation.

correlates inversely with the size of the cation[31]. Therefore, for the transfer of cations into 2DPP-OD-HEX ($DN_{Poly} \ll DN_{sol}$), the insertion of smaller cations entails a larger energy penalty. Hence, a strong cation size-dependent ion insertion is expected for the electrochemical doping of 2DPP-OD-HEX. Further, since $AN_{Poly} \approx 0 \ll AN_{Water}$, size-dependent ion insertion is also expected for anions. In short, charge-polarity dependent asymmetry in ion insertion should not be observed for 2DPP-OD-HEX. To verify this, we performed electrochemical doping experiments using 2DPP-OD-HEX as the OECT channel material.

P-OECT and n-OECT characteristics of 2DPP-OD-HEX OECTs were obtained in electrolyte solutions containing the same set of anions and cations and the same experimental configuration as was used for 2DPP-OD-TEG OECT measurements. For each ion, transfer characteristics were obtained on at least five different OECT devices to quantify variability (see Supplementary Fig. 17 showing all measured transfer characteristics). The mean value of $I_{D,max}$ measured for anion insertion into 2DPP-OD-HEX is observed to increase with increasing ion size (Fig. 5b) from tens of nA for F$^-$ to ~ -4 µA for ClO$_4^-$. This is like the ion size dependence observed with p-type 2DPP-OD-TEG OECTs (Fig. 2b).

For cation insertion, OECT devices gated through electrolytes containing Li$^+$ did not show turn ON upto $V_{G,REF}$ = 1.2 V (see Supplementary Fig. 17). For the other cations, mean $I_{D,max}$ increased with increasing cation size (Fig. 5b) from ~4 nA for Na$^+$ to ~80 nA for Cs$^+$ at $V_{G,REF}$ = 1.2 V. N-OECT measurements were also carried out in a 0.1 M aqueous solution of tetrabutylammonium chloride (TBACl). TBA$^+$ cation is relatively large with an ionic radius of 4.13 Å and n-type electrochemical doping resulted in even higher mean $I_{D,max}$ of ~1 µA at $V_{G,REF}$ = 1.1 V. Further, using CV measurements (see Supplementary Fig. 18), we ascertained that it is ion insertion that limits conductivity in 2DPP-OD-HEX used for OECT measurements.

The observed ion size dependence for both cation and anion insertion during electrochemical doping in 2DPP-OD-HEX is consistent with the proposed Gutmann DN/AN framework. The alkyl side-chains give the 2DPP-OD-HEX matrix an effectively low $DN_{Poly}$ and $AN_{Poly}$ i.e., the matrix is poorly coordinating toward both cations and anions. The large difference in both AN and DN between the polymer and water

leads to ion insertion being sensitive to $a_{A^+}$ and $b_{B^-}$, which are in turn cation and anion size-dependent, respectively.

## Donor/acceptor number framework for designing π-CPs

In DPP-based polymers reported in earlier works, glycol-based side-chains comprise up to 55% of the weight of the polymer (see Supplementary Table 3)[40–43]. These reports follow the traditional OECT materials design approach of utilizing hydrophilic glycol-based side-chains to enable passive solvent uptake by the polymer matrix with the aim of lowering the energetic barrier for ion-insertion. While passive swelling does enable facile ion insertion it also leads to operational instability and an unavoidable trade-off between ionic and electronic transport due to disruption of electronic pathways[14]. We note that none of these earlier reported DPP-based π-CPs showed n-type OECT operation despite DPP-backbone polymers generally reporting a high n-type mobility in the FET geometry[44] which is possibly an indication of the detrimental effect of passive swelling on electronic mobility.

N-type dopant densities at −1 V calculated from CVs range from $1.2 \times 10^{20}$ to $2.1 \times 10^{20}$ for the different cations used in this study. A side-chain number density of the same order is sufficient to interact with and stabilize the dopant ions in the polymer matrix. For example, in 2DPP-OD-TEG used in this study, TEG side-chains make up ~20% of the total weight of the polymer. This translates to ~$8 \times 10^{20}$ TEG side-chains per cm³ which corresponds to 4-8 TEG side-chains per cation (For these calculations, molecular weight of polymer repeat unit (Fig. 1a) is taken as 1480.19 g/mol and the density of the polymer is assumed to be 1 g/cm³). Thus, 20 wt% TEG side chains in 2DPP-OD-TEG is a sufficiently large side-chain density to enable dopant-ion stabilization via side-chain coordination.

To further verify this argument, we measured passive swelling of 2DPP-OD-TEG films. To quantify passive swelling, we measured the thickness of the polymer films in the dry state and under 0.1 M NaCl electrolyte[45]. Mean step heights of dry film and film measured under 0.1 M NaCl are 28.8 ± 2.8 nm and 22.2 ± 2.5 respectively. There was no evidence for passive swelling (see SI section 12). Thus, by having sufficient side-chain density to coordinate and stabilize cations but low enough side-chain density to minimize detrimental passive swelling, n-type doping with small cations is demonstrated in this work. A judicious choice of side-chain composition in π-CPs, such as for example, a mix of high DN/AN side-chain groups and hydrophobic alkyl side-chains[46], would favorably assist ion insertion and electrochemical doping while avoiding poor ionic/electronic mobilities and operational instability associated with passive solvent uptake.

Based on the DN/AN framework proposed here, we identify side-chain functional groups for tuning ion insertion energetics beyond the conventionally used glycol-side-chains[47]. Fig. 6 shows the DN and AN values of selected solvent molecules sorted according to their functional groups (see Supplementary Table 4). Note that cyclic or branched alkyl molecules are not included. We predict that π-CPs with

hydroxy (-OH)[11], amide (-N-(C = O)-)[48] or carboxylic acid (O = C-OH) based side-chain functional groups are expected to efficiently coordinate both cations and anions for high performance ambipolar mixed ionic electronic conductors. These groups have DN and AN values that are much higher than glycol side-chains (represented as ethers in Fig. 6) that are widely used for π-CPs with mixed ionic electronic conduction[49].

We anticipate that π-CPs functionalized with side-chains whose DN and AN are close to that of water could enable aqueous operable ambipolar electrochemical devices including single-component OECT based logic devices[50], dual-ion batteries[51] and electrochemically-active elements for neuromorphic computing[52]. The exact DN/AN values may be chosen based on the polymer and the dopant ion, keeping in mind that extremely strong ion-side-chain interactions might lead to ion trapping effects[53]. In general, the DN/AN framework provides a powerful tool to choose appropriate side-chain functional groups for high performance mixed ionic electronic conductors. It would also be exciting to study the suitability of this framework to rationalize and perhaps improve doping efficiencies in molecularly-doped organic semiconductors[54].

## Methods

2DPP-OD-TEG was synthesized using a Suzuki coupling polymerization between 3,6-bis(5-(4,4,5,5-tetramethyl-1,3,2-dioxaborolan-2-yl)thiophen-2-yl)-N,N-bis(2-octyldodecyl)-1,4-dioxopyrrolo[3,4-c]pyrrole and 3,6-bis(5-bromothiophen-2-yl)-N,N-bis(2-(2-(2-methoxyethoxy)ethoxy) ethyl)-1,4dioxopyrrolo[3,4-c]pyrrole. 2DPP-OD-HEX was synthesized using a Suzuki coupling polymerization between 3,6-bis(5-(4,4,5,5-tetramethyl-1,3,2-dioxaborolan-2-yl)thiophen-2-yl)-N,N-bis(2-octyldodecyl)-1,4-dioxopyrrolo[3,4-c]pyrrole, and 3,6-bis(5-bromothiophen-2-yl)-N,N-dihexyl-1,4-dioxopyrrolo[3,4-c]pyrrole. The reaction was carried out in the presence of $Pd_2(DBA)_3$ (DBA = dibenzylideneacetone) catalyst and the active ligand tri(o-tolyl)phosphine (P(o-tol)₃). Purification was done by precipitation in acetone followed by Soxhlet extraction using methanol, acetone, and hexane[17,55].

OECT chips with 25 identical OECT devices each with channel dimensions (width x length) = 100 μm × 10 μm were fabricated by photolithographically defining platinum metal contacts on SiO₂/Si substrates. An insulating layer of SU8-2002 (~2 μm thickness) was photolithographically patterned to insulate electrical contact lines from the electrolyte solution and to define individual devices. 2DPP-OD-TEG OECTs were prepared by spin coating a 10 mg/mL solution of 2DPP-OD-TEG in a 60:40 v/v mixture of 1,1,2,2-tetrachloroethane and chloroform (S D Fine Chemicals) on the OECT chip. The polymer solution mixture was stirred for 1 hour on a hot plate at 40 °C to dissolve the polymers completely before spin coating. The OECT chip was assembled in a custom built cell which confines the electrolyte above the chip[50]. A bare silver wire dipped in the electrolyte serves as the gate electrode and provides a return path for electrochemical current associated with the polymer redox. The gate voltage ($V_{G,REF}$) is measured across the OECT's source electrode, and an Ag/AgCl reference electrode dipped in the same electrolyte (see Fig. 1b).

Electrical measurements were carried out using a Keithley 2636B two-channel source-measure unit (SMU) interfaced with a computer using a custom computer program. The SMU channel used for applying the drain-source bias was configured in the 2-wire mode. The SMU channel for applying the gate bias was configured in 4-wire sense mode. The "sense-high" terminal of the gate bias SMU was connected to an Ag/AgCl reference electrode, while the "force-high" terminal was connected to the silver wire coil gate electrode.

Cyclic voltammetry experiments were performed using 2DPP-OD-TEG films (thickness ~57 nm) spin-coated on ITO/Glass substrates as working electrode in a custom-built CV cell in a 3-electrode configuration with the polymer (coated on ITO/Glass substrates) as the working electrode, an Ag/AgCl reference electrode (1 M NaCl fill

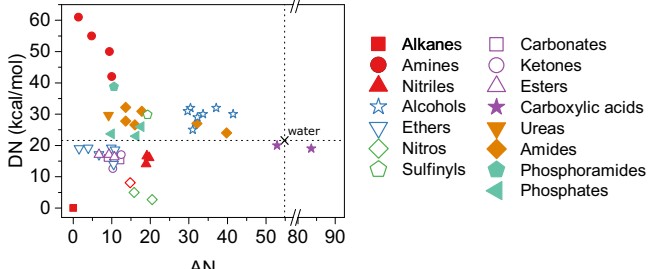

**Fig. 6 | DN and AN values of selected solvents sorted according to their functional groups.** Each family of molecules represents the variation of DN and AN values across different alkyl group sizes. The dotted horizontal and vertical lines represent the DN and AN values of water.

solution), and platinum wire coil as the counter electrode. The platinum wire counter is coiled around the reference electrode body (glass) and the tip of the reference electrode is kept at a distance of ~ 1 mm from the surface of the working electrode after cell assembly. The working electrode is sealed against the cell body using an O-ring exposing ~ 0.38 cm² of the film to the electrolyte. Nitrogen gas was bubbled through the electrolyte (~1 mL) during CV experiments. The cell design is shown in Supplementary Fig. 5.

Electrochemical Impedance spectroscopy (EIS) experiments were performed using the same cell configuration used for CV. EIS spectra were obtained over frequencies ranging from 10 kHz to 0.1 Hz at different potentials. Effective capacitance was calculated from the reactive component of the impedance using the equation $C_{eff} = -1/(2\pi f Z_{im})$[13,56]. $C_{eff}$ at 0.1 Hz is normalized to the volume of the film (area = 0.38 cm² and thickness ~57 nm) to obtain an effective volumetric capacitance $C^*_{eff}$.

Spectroelectrochemistry measurements were performed using a UV-Vis-NIR spectrophotometer (Ocean Insight) coupled with a Keithley 2636B SMU interfaced using a custom computer program. 2DPP-OD-TEG thin-film electrodes (~200 nm) were drop casted on 0.7 mm thick ITO/glass substrates. The electrode was assembled inside a 1 mm path-length quartz cuvette along with an Ag/AgCl reference electrode (1 M NaCl fill solution) and platinum wire counter electrode. For studying spectral signature changes during anion insertion, the potential applied to the polymer/ ITO/glass electrode was scanned from 0 V to 1 V in steps of 10 mV with a 500 ms hold time at each step. UV-vis-NIR absorption spectra were collected in intervals of 50 mV during the voltage scan. For studying n-type electrochemical doping, the working electrode potential was scanned from 0 V to −1.1 V.

## Data availability
Data that support the findings of this study are available from corresponding author upon reasonable request.

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

## Acknowledgements

J.J.S. thanks Deepika Jangid for help with CV experiments and feedback on the manuscript. Thanks to Karrothu Varun Kumar and Shivam Chopra for help with some of the preliminary experiments. Thanks to Dr. Surbhi Gupta for help with device fabrication. The authors thank the National Nanofabrication Center (NNfC) and Micro and Nano Characterization Facility (MNCF) at the Indian Institute of Science for fabrication and AFM characterization facilities. N.B.A acknowledges the new faculty start-up grant no. 120205-0618-77 provided by the Indian Institute of Science. This project is partially supported by the DST-IISc Energy Storage Platform on Supercapacitors and Power Dense Devices through the MECSP-2K17 program under grant no. DST/TMD/MECSP/2K17/20. S.P. and N.B.A. thank the Science and Engineering Research Board (SERB) for funding under the scheme of IRHPA grant no. IPA/2020/000033.

## Author contributions

J.J.S. and N.B.A. designed the experiments. J.J.S. and A.G. performed OECT, CV, and EIS experiments. S.P. conceived the idea of design and synthesis of polymers, provided input while drafting the manuscript. C.G. synthesized the polymers. S.R.M. performed AFM measurements. J.J.S. and N.B.A. wrote the manuscript. All authors reviewed and commented on the manuscript. N.B.A. supervised the project.

## Competing interests

The authors declare no competing interests.
