## [Peer review file · Nature Communications]

REVIEWER COMMENTS

Reviewer #1 (Remarks to the Author):

This manuscript was fascinating to read and very thought provoking. Great work!

Further comments attached.

Samuel and coworkers study the ion insertion of different cations and anions into a glycolated side chain conjugated polymer. They observe the previously reported anion size dependence. Next, they observe that cation insertion is largely independent of ion size in this polymer. They propose an electrolyte theory framework to explain the ion dependent effects in OECTs. They validate this framework by comparing the results of a fully alkyl version of the polymer and show that in that version of the polymer cation insertion does depend on ion size as their theory predicts. The proposed use of Gutman donor/acceptor numbers as guidelines for design of new/better mixed conductors represents a valuable contribution to the field and certainly warrants further investigation. Overall, the paper is very well written and complete, with strong evidence to support its claims. For this reason, I suggest publication of this manuscript in *Nature Communication* following minor revisions as outlined below.

Before publication the authors should address the following:

- 1) The polarizability of the cations used in the main text are very similar. Inclusion of a polyatomic cation would create a more convincing argument that cation insertion is truly ion size/polarizability independent. The TBA⁺ data shown in the SI convincingly demonstrates that cation insertion is independent of cation properties in the TEG polymer and would be valuable included in the main text, or at least mentioned directly in the discussion around Figure 3.
- 2) The p-type operation of this material extends well beyond the electrochemical stability of water. It is likely there is significant formation of hydrogen peroxide¹ and or acidification² of the solution. Control experiments looking for the presence of these species after cycling would be valuable. These unwanted side reactions could explain the current observed in the CVs at high potential in Figure 2C for the F and Cl anions.
- 3) The lack of p-type operation in Cl⁻ is not surprising for this polymer (end of page 4). Based on the CVs in acetonitrile/TBAPF₆ in this study the HOMO level for this polymer is -5.25 eV. With the known ~200 mV shift in threshold voltage when changing from PF₆⁻ to Cl⁻ supporting electrolyte,³ and the nearness to the oxygen reduction reactions (mentioned above) the fact that the p-OECT does not turn on in F⁻ or Cl⁻ is expected and presenting that result as a surprise weakens the overall argument.
- 4) Many of the figures in the SI show more than one curve on the same plot without a legend. A description of what each curve corresponds to (different cycle number, different device, or different doping potential) would help clarify the meaning of the plots. (Figure S3, S6, S7, S9, S11 specifically).

References:

- (1) Giovannitti, A.; Rashid, R. B.; Thiburce, Q.; Paulsen, B. D.; Cendra, C.; Thorley, K.; Moia, D.; Mefford, J. T.; Hanifi, D.; Weiyuan, D.; Moser, M.; Salleo, A.; Nelson, J.; McCulloch, I.; Rivnay, J. Energetic Control of Redox-Active Polymers toward Safe Organic Bioelectronic Materials. *Adv. Mater.* **2020**, *32* (16), 1–9. <https://doi.org/10.1002/adma.201908047>.
- (2) Cendra, C.; Giovannitti, A.; Savva, A.; Venkatraman, V.; McCulloch, I.; Salleo, A.; Inal, S.; Rivnay, J. Role of the Anion on the Transport and Structure of Organic Mixed Conductors. *Adv. Funct. Mater.* **2018**, *29*, 1807034.
- (3) Flagg, L. Q.; Bischak, C. G.; Onorato, J. W.; Rashid, R. B.; Luscombe, C. K.; Ginger, D. S. Polymer Crystallinity Controls Water Uptake in Glycol Side-Chain Polymer Organic Electrochemical Transistors. *J. Am. Chem. Soc.* **2019**, *141* (10), 4345–4354. <https://doi.org/10.1021/jacs.8b12640>.

Reviewer #2 (Remarks to the Author):

The manuscript by Samuel et al. proposes to use the Gutmann donor/acceptor number framework to qualitatively explain the charge-polarity-dependent ion insertion asymmetry observed for the ambipolar diketopyrrolopyrrole-based polymer 2DPP-OD-TEG. More specifically, the authors observed that anion insertion during electrochemical doping of 2DPP-OD-TEG strongly correlates with the anion size while cation insertion is size-independent. Several studies have investigated the ion size-dependent OECT response of mixed ion-electron conducting polymers with and without glycolated sidechains (e.g., <https://doi.org/10.1021/acs.chemmater.0c01984>, <https://doi.org/10.1002/adfm.201807034>, <https://doi.org/10.1021/acs.chemmater.8b02220>). Thus, a mechanistic interpretation of the phenomena responsible for ion insertion in mixed ion-electron conducting polymers is certainly timely. While this study provides some insight into the role of hydrophilic side chains on ion intercalation and transport in this class of polymers, it seems to oversimplify the underlying physics. Below is a list of major concerns:

- 1) The authors only consider the size of the ions, while the hardness/softness, as well as the hydrophilicity/hydrophobicity nature of the ions, is not discussed. Anions such as ClO_4^- and TFSI^- are more hydrophobic than the smaller F^- and Cl^- and are known to have a higher affinity with polymers carrying hydrophilic side chains.
- 2) On the technical side, why did the authors use platinum as the source/drain electrode material instead of the widely used gold? How do the different electrode materials affect the OECT response?
- 3) On page 4, the authors write “the frontier energy levels of the polymer are comparable to that of water, suggesting that polymer oxidation (corresponding to p-doping) should be possible”. This sentence is unclear, and it seems to suggest that water oxidizes the polymer.
- 4) As stated by the authors, the TEG sidechains make up only ~20% of the total weight of the polymer. Therefore, it is hard to understand how this small portion of the polymer chain can significantly impact ion insertion.
- 5) On page 17, it is stated that the polymer crystallinity or structure does not play a significant role in the observed charge polarity-dependent ion insertion. How was this proven? 2DPP-OD-HEX and 2DPP-OD-TEG could have very different mesostructures.
- 6) Passive swelling is assessed by measuring the film thickness changes before and after water immersion. This way of measuring passive swelling is inaccurate as the films are expected to collapse to

their initial state once dried. So, this part is very little informative and does not support the claim that 2DPP-OD-TEG does not passively swell.

7) While Figure 2b suggests that I_d increases with the ion size, the CV data reported in Figure 2c do not correlate with this statement. Why? Also, I found it puzzling that NO_3^- yields one order of magnitude larger current than F^- and Cl^- , but gives almost identical effective capacitance to these two small anions. Why?

8) The authors write that V_g was swept to values up to which stable OECT operation was observed. What happens at higher gate voltage values?

Reviewer #3 (Remarks to the Author):

01. The manuscript lacks novelty to be published in Nature Communications. Side chain dependent electrochemical doping of conjugated polymers has been studied extensively over the past years, especially in OECTs, with similar observations and conclusions.

02. The proposed hypothesis lacks generality.

While the Gutman donor and acceptor numbers provide a qualitative explanation for the doping behavior of the two DPP polymers presented in the study, the hypothesis has not been validated with other functional groups or polymers.

03. The manuscript fails to describe how DN and AN of polymers can be quantified, especially for copolymers with different side chains.

For example, in the case of the 2DPP-OD-TEG polymer, TEG side chains make up just 20% of the total polymer weight, and the cation coordinating power (DN) is said to be similar to that of water. How much can the TEG : Alkyl side chain ratio be tuned to obtain different ion-dependent doping properties?

04. Polymer dopability is argued in terms of how well the ions can be coordinated by the side chains compared to solvation by the electrolyte. What are the possibilities of solvated ions entering the polymer during electrochemical doping?

05. Designing polymers with high DN/AN side chains might not always be beneficial as mentioned in the manuscript. Strong ion attraction can cause ion trapping and limit the reversibility of the doping process.

Title: Charge Polarity-Dependent Ion-Insertion Asymmetry During Electrochemical Doping of an Ambipolar π -Conjugated Polymer

Authors: Jibin J Samuel, Ashutosh Garudapalli, Chandrasekhar Gangadharappa, Smruti Rekha Mahapatra, Satish Patil, Naga Phani B. Aetukuri*

Response to Reviewer Comments

We thank the reviewers for their feedback and interest in our work. We reproduced the reviewer's comments in blue colored font and our responses to the comments in normal type font. The changes made in the revised manuscript in response to review comments are highlighted in yellow. The comments are listed and addressed point-by-point below:

Reviewer 1 remarks:

Samuel and coworkers study the ion insertion of different cations and anions into a glycolated side chain conjugated polymer. They observe the previously reported anion size dependence. Next, they observe that cation insertion is largely independent of ion size in this polymer. They propose an electrolyte theory framework to explain the ion dependent effects in OECTs. They validate this framework by comparing the results of a fully alkyl version of the polymer and show that in that version of the polymer cation insertion does depend on ion size as their theory predicts. The proposed the use of Gutman donor/acceptor numbers as guidelines for design of new/better mixed conductors represents a valuable contribution to the field and certainly warrants further investigation. Overall, the paper is very well written and complete, with strong evidence to support its claims. For this reason, I suggest publication of this manuscript in Nature Communication following minor revisions as outlined below.

Before publication the authors should address the following:

We thank the reviewer for their interest and positive assessment of our work.

1. The polarizability of the cations used in the main text are very similar. Inclusion of a polyatomic cation would create a more convincing argument that cation insertion is truly ion size/polarizability independent. The TBA^+ data shown in the SI convincingly demonstrates that cation insertion is independent of cation properties in the TEG polymer and would be valuable included in the main text, or at least mentioned directly in the discussion around Figure 3.

We thank the reviewer for this suggestion. In order to strengthen our arguments, in line with the reviewer's suggestions, we added a discussion regarding n-type doping with TBA^+ in page 10 of the revised main text. We have reproduced the text below for convenience:

“All the cations used are monoatomic cations with comparable polarizabilities. To confirm that n-type electrochemical doping in 2DPP-OD-TEG is indeed not dependent on ion-size, OECT, CV, EIS experiments were performed with a large polyatomic cation, tetra butyl ammonium (TBA^+) with an ionic radius of 4.13 \AA .²⁴ OECTs gated through 0.1 M TBACl electrolyte solution showed n-type operation with an $I_{D,\text{max}} = 462 \pm 165 \text{ nA}$ (at $V_{G,\text{REF}} = 0.9 \text{ V}$) and an OECT turn-on voltage of $V_{T,n} = 0.69 \pm 0.03 \text{ V}$ (also see SI section 8 and Supplementary Figs. 14a-c). The mean charge under the oxidative wave of CVs is 146 \mu C corresponding to a doping density of $\sim 4.2 \times 10^{20} \text{ carriers/cm}^3$ at -1 V And, C_{eff}^* estimated from EIS measurements performed at -1 V is 136 F/cm^3 (Supplementary Fig. 14d). The charge density and C_{eff}^* values are of the same order as for other cations studied in this work suggesting that size independent electrochemical doping is observed even for TBA^+ , a polyatomic cation.

2. The p-type operation of this material extends well beyond the electrochemical stability of water. It is likely there is significant formation of hydrogen peroxide¹ and or acidification² of the solution. Control experiments looking for the presence of these species after cycling would be valuable. These unwanted side reactions could explain the current observed in the CVs at high potential in Figure 2C for the F and Cl anions.

As the reviewer pointed out, the potentials used for OECT and CV experiments for p-type electrochemical doping lie outside the electrochemical stability window of water. This corresponds to the range of -0.649 V to $+0.580 \text{ V}$ vs. Ag/AgCl (1M NaCl, fill solution) at pH = 7. At the potentials of 1 to 1.2 V vs. Ag/AgCl used in p-type electrochemical doping experiments, the parasitic reactions that occur are oxygen evolution reactions due to water and

H₂O₂ decomposition (D. M. de Leeuw et al., Synthetic Metals, 87 (1997) 53-59). However, the deep HOMO level of 2DPP-OD-TEG at 5.25 eV implies H₂O₂ generation by oxygen reduction in ambient conditions is suppressed (A. Giovannitti et al., Adv. Mater. 2020, 32, 1908047). As a result, the reaction involving O₂ evolution from H₂O₂ is not a possibility. The relevant parasitic reaction is the decomposition of H₂O evolving O₂ and generating H⁺ which might result in acidification.

An alternate possibility that could have led to the low p-type doping for F⁻ and Cl⁻ insertion could be the simultaneous de-doping of the polymer through the reaction: $4\{P^+:B_{Poly}^-\} + 2H_2O \rightleftharpoons 4P + 4B_{Sol}^- + O_2 + 4H^+$. The following figure summarizes the possible parasitic reactions during p-type doping. We attempted to quantify parasitic reactions using CV experiments, as discussed below.

We performed CVs up to an upper vertex potential corresponding to the maximum gate voltage in OECT experiments for each of the anions used in this study (see Supplementary Fig. 7). An upper bound on the absolute amount of charge consumed by parasitic reactions per cycle

(ΔQ_{Irrev}) is calculated from the difference between the integrated charge under the oxidative (Q_{Ox}) and reductive (Q_{Red}) waves in the CV curve ($\Delta Q_{Irrev} = Q_{Ox} - Q_{Red}$). ΔQ_{Irrev} is <10 μC even at 1.2 V in NaCl and NaF electrolytes (see Supplementary Fig. 7e). This corresponds to ~ 0.1 nano mole of the parasitic side-products generated per cycle (electrolyte volume is 1 mL for CV experiments in this study). 0.1 nanomole is too small to be quantified accurately with standard pH measurements around $\text{pH} = 7$. pH measurements of the electrolyte before and after holding at the potential at $V_{CV,max}$ for 4 hours did not show any measurable change in pH. Furthermore, the maximum oxidative charge (Q_{Ox}) for both F^- and Cl^- ion-insertion at 1.2 V are 25 μC and 14 μC , which is far below the Q_{Ox} for ClO_4^- and NO_3^- (95 μC and 42 μC). Notably, de-doping through the parasitic pathway is insufficient to explain the low charge insertion for electrochemical doping with F^- and Cl^- . Therefore, the central conclusions from p-type doping experiments which reflect a strong ion-size dependence are unaffected by the presence of parasitic reactions, if any.

We added a discussion in SI section 3, page 6 discussing this in detail. The text and the added figures are reproduced below for convenience.

“The potentials used for OECT and CV experiments for p-type electrochemical doping lie outside the electrochemical stability window of water which correspond to the range of -0.649 V to +0.580 V vs. Ag/AgCl (1M NaCl fill solution) at $\text{pH} = 7$. At oxidative potentials of the order of 1 to 1.2 V vs. Ag/AgCl used for the p-type electrochemical doping experiments, the oxygen reduction reactions (ORR)² given by equations 1 and 2 (redox potentials referenced to Ag/AgCl with 1M NaCl fill solution) occur in the backward direction resulting in O_2 evolution:

The redox reaction involved in p-type doping is given by equation 3.

Since the reaction, $2\text{P} + \text{O}_2 + 2\text{H}^+ + 2\text{e}^- \rightleftharpoons 2\text{P}^+ + \text{H}_2\text{O}_2$ has $E_{\text{redox}} = 0.033 - 0.58 = -0.55$ $\text{V} < 0$, it is not feasible thermodynamically. In other words, the deep HOMO level of 2DPP-OD-TEG at -5.25 eV implies H_2O_2 generation by ORR in ambient conditions is suppressed (see Supplementary Fig. 6).³As a result, the reaction involving O_2 evolution from H_2O_2 can be

discounted. The relevant parasitic reaction during p-type electrochemical doping of 2DPP-OD-TEG is therefore decomposition of H₂O evolving O₂ and generating H⁺ which might result in acidification of the electrolyte. However, the kinetic overpotentials for water oxidation on polymeric surfaces are known to be high.² Therefore, the fraction of charge that leads to the acidification of the electrolyte is expected to be small.

The parasitic water decomposition/oxygen evolution reaction can cause de-doping of the channel through the reaction given by equation 4 (also see Supplementary Fig. 6).

To ensure that this is not the reason for lack of p-type doping with smaller anions such as F⁻ and Cl⁻, we performed CV experiments in different anion containing electrolytes to quantify the parasitic reactions. CVs were measured on 2DPP-OD-TEG films at 100 mV/s scan rate from 0.2 V to an upper vertex potential corresponding to the maximum OECT gate voltages (see Supplementary Figs. 7 a-d).

An upper bound on the absolute amount of electrochemically generated parasitic side-products per cycle (ΔQ_{Irrev}) is calculated from the difference between the integrated charge under the oxidative (Q_{Ox}) and reductive (Q_{Red}) waves in the CV curve ($\Delta Q_{Irrev} = Q_{Ox} - Q_{Red}$). The measured ΔQ_{Irrev} is <10 μ C even at an oxidation potential of 1.2 V in NaCl and NaF electrolytes (see Supplementary Fig. 7e). This corresponds to ~0.1 nano mole of the parasitic side-products generated per cycle (the volume of the electrolyte used in OECT experiments is ~ 1ml) . The corresponding change in pH due to the acidification of the electrolyte, if any, is insignificant around pH = 7. Further, ΔQ_{Irrev} is a small fraction of the total oxidative charge deposited during CV (see Supplementary Fig. 7f) and absolute value of ΔQ_{Irrev} for the larger anions is higher than for the smaller ions (see Supplementary Fig. 7e). Furthermore, the maximum oxidative charge (Q_{Ox}) for both F⁻ and Cl⁻ ion-insertion at 1.2 V are 25 μ C and 14 μ C, which is far below the Q_{Ox} of 95 μ C and 42 μ C respectively for ClO₄⁻ and NO₃⁻ at 1 V and 1.05 V. Notably, de-doping as a parasitic pathway is insufficient to explain the low charge insertion for electrochemical doping with F⁻ and Cl⁻. This implies that the parasitic side-reactions do not play any significant role in the observed ion-size dependence during p-type doping of 2DPP-OD-TEG.

Supplementary Figure 1. Schematic showing the hydrogen evolution and oxygen evolution reactions (ORR) along with polymer Ionization Potential (IP) and Electron Affinity (EA) values. The neutral state of the polymer is stable against H_2O_2 -generating ORR reaction because of the deep HOMO level of 2DPP-OD-TEG. However, the p-doped state can be dedoped by the parasitic reactions involving oxidation of water to O_2 .

Supplementary Figure S7. (a-d) 5 cycles of cyclic voltammograms at 100 mV/s in electrolytes containing different anions going up to increasing maximum potentials, $V_{CV,max}$ (e) Mean irreversible charge, ΔQ_{irrev} (corresponding to parasitic electrochemical reactions) as a function of $V_{CV,max}$ calculated from the difference between the charge under oxidative and reductive waves. $\Delta Q_{irrev} \sim 7, 8, 15, 12, 2 \mu C$ for F^- , Cl^- , NO_3^- , ClO_4^- and blank ITO (in NaCl) respectively at the highest used $V_{CV,max}$. (f) Percentage of ΔQ_{irrev} relative to Q_{ox} , the mean oxidative charge deposited during the oxidative wave of the CV, as a function of $V_{CV,max}$.

We also added the following discussion in page 7 of main text of the revised manuscript.

“A quantitative estimate of the volumetric density of inserted ions can be obtained from the charge transferred during the CV scans. The current in the oxidative wave comprises of currents arising out of reversible processes such as charging of the electrostatic double layer (EDL) at the polymer/electrolyte interface and ion insertion into/oxidation of the bulk of the polymer film and irreversible processes such as parasitic side reactions (e.g., electrolyte decomposition – O₂ evolution). During the reductive scan, only the charge corresponding to reversible processes is recovered and this provides a lower bound for the number of ions that were originally inserted into the polymer film. The magnitude and fraction of the irreversible charge is quantified as a function of the CV upper vertex potential for all the ions (see Supplementary Fig. 7e,f). Parasitic reactions can reduce and de-dope the polymer film but form only a small fraction of the total charge and hence are not expected to significantly affect the ion-insertion and doping processes. Therefore, the central conclusions of ion-size dependence are unaffected by parasitic reactions, if any (also see SI section 3).”

3. The lack of p-type operation in Cl⁻ is not surprising for this polymer (end of page 4). Based on the CVs in acetonitrile/TBAPF₆ in this study the HOMO level for this polymer is -5.25 eV. With the known ~200 mV shift in threshold voltage when changing from PF₆⁻ to Cl⁻ supporting electrolyte,³ and the nearness to the oxygen reduction reactions (mentioned above) the fact that the p-OECT does not turn on in F⁻ or Cl⁻ is expected and presenting that result as a surprise weakens the overall argument.

While we acknowledge that ORR and ion-insertion could be competing electrochemical reactions at the potentials of interest for these experiments, we disagree that the shift in threshold voltage would be 200 mV in our experiments. This is because, the total shift in the threshold voltage across all cation sizes measured is <50 mV and the estimated shift in the threshold voltage across anions is ~100 mV (see Supplementary Fig. 4c). Therefore, the lack of Cl⁻ or F⁻ ion insertion need not be affected by the competing water decomposition reaction. Furthermore, we note that the report by Flagg et al. on ion-size dependence in p-type doping explains the size dependence in terms of the ease with which different ions shed their hydration shells and insert into the hydrophobic polymer matrix. Based on the conclusions of their work, two ions with different charges but comparable hydration energies should not have different doping abilities. However, based on the results of our work, Na⁺ and Cl⁻ with comparable

hydration energies show starkly different insertion tendency with Na^+ insertion being possible, but Cl^- insertion being negligibly small. Therefore, the absence of Cl^- is surprising based on our extant understanding of electrochemical doping. Clearly, an alternate mechanism to contextualize these results is needed.

We updated the text in page 4 of the main manuscript to reflect this:

“The lack of p-type OECT operation is surprising considering the following factors together. 1) 2DPP-OD-TEG shows ambipolar conduction in organic field effect transistor (OFET) geometry¹⁷ which implies sufficient mobilities of electrons and holes, 2) the frontier energy levels of the polymer (Supplementary Information (SI) section 1, Supplementary Table 1) are comparable to the electrochemical stability window of the aqueous electrolyte which implies that both the n- and p-doped states must be stable despite contact with the electrolyte and 3) Cl^- ions have a lower hydration energy (-344 kJ/mol) compared to Na^+ (-383kJ/mol)¹⁸ which implies it should have been easier for Cl^- ions to insert into the polymer matrix to dope it p-type.¹⁹”

4. Many of the figures in the SI show more than one curve on the same plot without a legend. A description of what each curve corresponds to (different cycle number, different device, or different doping potential) would help clarify the meaning of the plots. (Figure S3, S6, S7, S9, S11 specifically).

We thank the reviewer for pointing this out. We have added legends to the figures and descriptions in the figure captions to clarify the meaning of the different colours used in the mentioned supplementary information figures. The updated figures in the revised manuscript are numbered S3, S8, S10, S12, S17 respectively.

Reviewer 2 remarks:

The manuscript by Samuel et al. proposes to use the Gutmann donor/acceptor number framework to qualitatively explain the charge-polarity-dependent ion insertion asymmetry observed for the ambipolar diketopyrrolopyrrole-based polymer 2DPP-OD-TEG. More specifically, the authors observed that anion insertion during electrochemical doping of 2DPP-OD-TEG strongly correlates with the anion size while cation insertion is size-independent. Several studies have investigated the ion size-dependent OECT response of mixed ion-electron

conducting polymers with and without glycolated sidechains (e.g., <https://doi.org/10.1021/acs.chemmater.0c01984>, <https://doi.org/10.1002/adfm.201807034>, <https://doi.org/10.1021/acs.chemmater.8b02220>). Thus, a mechanistic interpretation of the phenomena responsible for ion insertion in mixed ion-electron conducting polymers is certainly timely. While this study provides some insight into the role of hydrophilic side chains on ion intercalation and transport in this class of polymers, it seems to oversimplify the underlying physics. Below is a list of major concerns:

We thank the reviewer for their interest in our work. We address the reviewer's comments below.

1) The authors only consider the size of the ions, while the hardness/softness, as well as the hydrophilicity/hydrophobicity nature of the ions, is not discussed. Anions such as ClO₄⁻ and TFSI⁻ are more hydrophobic than the smaller F⁻ and Cl⁻ and are known to have a higher affinity with polymers carrying hydrophilic side chains.

The concept of hydrophilicity of an ion or the affinity of the ion for water can be quantified in terms of the free energy of hydration (ΔG_{hyd}) of the ion. The free energy of hydration is inversely correlated to size of the ion (Y. Marcus, J. Chem. Soc. Faraday Trans., 1991, Vol. 87). The hardness of the ion is also correlated with ionic size – hardness decreases with increasing ionic size for both cations and anions. Hardness of the ion can be quantified by η , the absolute chemical hardness proposed by Pearson (R. G. Pearson, Inorg. Chem. 1988, 27, 4, 734–740). Thus, effects of both hydrophilicity/hydrophobicity and hardness/softness, being correlated with ion size, are also (indirectly) included in our ion-size dependence study. However, we note that there could be differences when the ions are not spherically symmetric. In our studies, all the ions were chosen to have spherical symmetry, therefore, the Lewis acidity/basicity correlates with ion size. We have added the following text regarding the same in page 10 in the main text of revised manuscript.

“As the ionic radii is correlated with hydration energy and hardness of the ion, our observations can also be interpreted in terms of hydration energy or the Pearson hardness. We further discuss this in SI section 7.”

We also added a discussion in section 7 of the revised SI (page 15), along with a plot of ion-size dependence of figures 2b and 3b in terms of ΔG_{hyd} and η in Supplementary Fig. 13 (reproduced below).

“The concept of hydrophilicity of an ion or the affinity of the ion for water can be quantified in terms of the free energy of hydration (ΔG_{hyd}) of the ion. ΔG_{hyd} is inversely correlated with the size of the ion.⁴ The hardness of the ion is also correlated with ionic size – hardness decreases with increasing ionic size for both cations and anions. Hardness is quantified by the Pearson hardness, η .⁵ Thus, effects of both hydrophilicity/hydrophobicity and hardness/softness, being correlated with ion size, are also indirectly accounted for in ion-size dependence. To further emphasize this, OEECT $I_{D,max}$ is replotted in terms of ΔG_{hyd} and η of the dopant ions in Supplementary Fig. 13.”

Supplementary Figure 13. Mean $I_{D,max}$ as a function of (a) free energy of hydration of ions⁴ and (b) Pearson hardness⁵. Pearson hardness numbers for ClO_4^- and NO_3^- are not readily available.

2) On the technical side, why did the authors use platinum as the source/drain electrode material instead of the widely used gold? How do the different electrode materials affect the OEECT response?

We found in our preliminary experiments that gold is not electrochemically stable at high oxidative potentials in the presence of chloride ions. We observed etching of gold contacts under the polymer film during p-type OEECT experiments in NaCl electrolyte, while platinum

was found to be stable. Therefore, for consistency, all the OECT experiments in this work were performed using OECTs with platinum metal contacts. We note that changing contact metal does not significantly change OECT parameters as evident from the below figure comparing typical OECT measurements on Au and Pt contact devices.

3) On page 4, the authors write “the frontier energy levels of the polymer are comparable to that of water, suggesting that polymer oxidation (corresponding to p-doping) should be possible”. This sentence is unclear, and it seems to suggest that water oxidizes the polymer.

We thank the reviewer for pointing this out. We have incorporated the appropriate changes in page 4 of the revised main text to clarify the intended meaning. The text is reproduced below for convenience.

“...2) the frontier energy levels of the polymer (Supplementary Information (SI) section 1, Supplementary Table 1) are comparable to the electrochemical stability window of the aqueous electrolyte, which implies that both the n- and p-doped states must be stable despite contact with the electrolyte ...”

4) As stated by the authors, the TEG sidechains make up only ~20% of the total weight of the polymer. Therefore, it is hard to understand how this small portion of the polymer chain can significantly impact ion insertion.

This is an interesting question. First, we would like to note that 20 wt% of TEG side chains of the polymer is a sufficiently large fraction for the dopant densities required during OECT operation. For example, the calculated volumetric density of inserted cations during n-type

doping is in the range of 1.2×10^{20} to 2.1×10^{20} cations per cm^3 of the polymer. By comparison 20 wt% of TEG chains correspond to $\sim 8 \times 10^{20}$ TEG chains per cm^3 of the polymer. In other words, there are 4-8 TEG side-chains for every dopant ion. This is a sufficiently large number for the coordination of dopant ions by the TEG side chains and stabilize the dopant ion in the polymer matrix making ion-insertion energetically feasible. This observation further strengthens our key arguments that dopant-ion side-chain interactions play an important role in ion-insertion during electrochemical doping in π -CPs. We added the following discussion to page 20 of the main text of the revised manuscript.

“N-type dopant densities at -1 V calculated from CVs range from 1.2×10^{20} to 2.1×10^{20} for the different cations used in this study. A side-chain number density of the same order is sufficient to interact with and stabilize the dopant ions in the polymer matrix. For example, in 2DPP-OD-TEG used in this study, TEG side-chains make up $\sim 20\%$ of the total weight of the polymer. This translates to $\sim 8 \times 10^{20}$ TEG side-chains per cm^3 which corresponds to 4-8 TEG side-chains per cation (For these calculations, molecular weight of polymer repeat unit (Fig 1a) is taken as 1480.19 g/mol and the density of the polymer is assumed to be $= 1 \text{ g/cm}^3$). Thus, 20 wt% TEG side chains in 2DPP-OD-TEG is a sufficiently large side-chain density to enable dopant-ion stabilization via side-chain coordination.”

5) On page 17, it is stated that the polymer crystallinity or structure does not play a significant role in the observed charge polarity-dependent ion insertion. How was this proven? 2DPP-OD-HEX and 2DPP-OD-TEG could have very different mesostructures.

We thank the reviewer for pointing this out. The discussion on polymer crystallinity is not required at this place in the text. We removed this sentence and have rewritten the section for the sake of better clarity. The changed text is in Page 19 of the main text in the revised manuscript.

“In DPP-based polymers reported in earlier works, glycol-based side-chains comprise up to 55% of the weight of the polymer (see Supplementary Table 3).³⁹⁻⁴² These reports follow the traditional OECT materials design approach of utilizing hydrophilic glycol-based side-chains to enable passive solvent uptake by the polymer matrix with the aim of lowering the energetic barrier for ion-insertion...”

6) Passive swelling is assessed by measuring the film thickness changes before and after water immersion. This way of measuring passive swelling is inaccurate as the films are expected to collapse to their initial state once dried. So, this part is very little informative and does not support the claim that 2DPP-OD-TEG does not passively swell.

We thank the reviewer for this question. In order to address the reviewer's concern that the soaked film may dry out and collapse to the initial state, we measured the film thickness in air and then under the electrolyte after allowing the film to soak for ~1 hour. We updated SI section 12 with the new set of results. Our repeat experiments do conclude that 2DPP-OD-TEG does not undergo significant passive swelling. We note that the films used for these experiments are from a new batch of polymer synthesis and therefore the dry film thicknesses were ~30 nm, smaller than the ~50 nm films that we were able to obtain with a previous batch of the polymer.

We have updated the experimental results in page 24 of the revised supplementary information. The added text and figures are reproduced below for convenience.

“Passive swelling of 2DPP-OD-TEG was assessed using atomic force microscopy (AFM). The polymer film were prepared for swelling studies by spin coating on ITO/glass. A thin scratch was made on the film using the point of a hypodermic needle. The film was first imaged across the scratch, in the dry state. A sample AFM image of the dry film is shown in Supplementary Fig. 19a. Next, the film was covered with aqueous 0.1 M NaCl solution and allowed to soak for 1 hour. Then AFM measurements were performed on the film (Supplementary Fig. 19b) while still in contact with the electrolyte using in-liquid imaging mode, to prevent any drying/shrinking of the film. Multiple measurements were performed and mean terrace heights were calculated using Gwyddion AFM analysis tool (Supplementary Fig. 19c).¹¹

Supplementary Figure 2. (a) AFM image across a scratch made on the dry polymer film (b) AFM image of the same film under 0.1 M aqueous NaCl solution after having soaked in the electrolyte for 1 hour. (c) Box plot showing the step heights calculated for multiple AFM scans of the dry film and the film under 0.1 M NaCl solution. Mean step heights of dry film and film measured under 0.1 M NaCl are 28.8 ± 2.8 nm and 22.2 ± 2.5 nm respectively

We also updated the discussion in the revised main text in page 20 as reproduced below:

“To quantify passive swelling, we measured the thickness of the polymer films in the dry state and under 0.1 M NaCl electrolyte.⁴⁴ Mean step heights of dry film and film measured under 0.1 M NaCl are 28.8 ± 2.8 nm and 22.2 ± 2.5 nm respectively. There was no evidence for passive swelling (see SI section 12).”

7) While Figure 2b suggests that I_d increases with the ion size, the CV data reported in Figure 2c do not correlate with this statement. Why? Also, I found it puzzling that NO_3^- yields one order of magnitude larger current than F^- and Cl^- , but gives almost identical effective capacitance to these two small anions. Why?

We thank the reviewer for this question. We note that the EIS measurements were all limited to a maximum oxidative potential of 1 V to minimize contributions from increasing parasitic currents at higher voltages. To be consistent with EIS measurements, the CVs were also likewise limited to 1 V. However, OECT measurements were carried out until turn ON or a maximum oxidative potential of 1.2 V for p-type doping. Therefore, a direct comparison between capacitance and channel currents was not straight forward. Note that all potentials are referenced against an Ag/AgCl electrode.

Therefore, to resolve the apparent mismatch between OECT and CV/EIS experiments, we repeated the CV experiments by extending maximum potential of the CV scan to correspond with the OECT experiments. We then calculated the charge recovered during the backward/reductive scan which gives a lower bound for the estimated density of inserted ions and hence the charge carrier concentration. The data from these experiments is now added to a revised Figure 2c and d in the main text (reproduced below for convenience).

It can be seen from these figures that the integrated CV charge density for NO_3^- and ClO_4^- is significantly higher than that observed for F^- and Cl^- doping. Specifically, for ClO_4^- , NO_3^- , the

charge under reductive wave of the CV are 84 μC (at $V_{\text{G,REF}} = 1 \text{ V}$) and 28 μC (at $V_{\text{G,REF}} = 1.05 \text{ V}$), respectively. By contrast, the estimated charge for F^- and Cl^- is 17 and 6 μC , respectively, which is within the typical range of charge stored in EDL and volumetric charging cannot be resolved. Clearly, the anion size-dependent volumetric capacitance and the channel currents are consistent. We hope that this new set of data addresses the reviewer's concern and further strengthens the central conclusions of this work. We updated Fig. 2 and the discussion in pages 7 and 8 in the revised main text as follows.

“Fig. 2d shows the mean value of the charge under the reductive wave of the CV averaged over 5 cycles. The corresponding doping density is shown on the right axis (assuming 100% doping efficiency). The dashed lines represent the equivalent charge that could be typically stored in a EDL at $V_{\text{CV,max}}$ for each of the electrolytes considering the upper limit of typical EDL capacitance of 40 $\mu\text{F}/\text{cm}^2$. In the case of F^- and Cl^- , the total charge under the reductive wave was calculated to be 17 and 6.3 μC . This is not distinguishable from the charge that could be stored in the EDL (18 μC at 1.2V, electrode area = 0.38 cm^2) at the polymer/electrolyte interface.

However, for NO_3^- and ClO_4^- , the total charge recovered on the reductive scan is significantly higher than that corresponding to EDL charging. This confirms that volumetric ion insertion does indeed take place in the case of NO_3^- and ClO_4^- . The charge under the reductive wave for ClO_4^- ions at 1 V is 84 μC which corresponds to a volumetric doping density of $2.4 \times 10^{20} / \text{cm}^3$. This value is higher than that for NO_3^- (28 μC or $8.0 \times 10^{19} / \text{cm}^3$) at 1.05 V. The doping density follows the trend observed for $I_{\text{D,max}}$ as a function of the dopant anions (Fig. 2b). However, we note that the relation is not linear. For example, the corresponding OECT $I_{\text{D,max}}$ in the case of ClO_4^- is 12 times higher than for NO_3^- change while the estimated densities of inserted ions are ~ 3 times higher. This is not surprising given the known super linear relation between doping density and conductivity in organic semiconductors.²²”

Figure 3. Anion size dependence for p-type electrochemical doping of 2DPP-OD-TEG (a) Typical p-type OECT transfer characteristics of 2DPP-OD-TEG OECTs measured in 0.1 M aqueous solutions of sodium salts with different anions as mentioned in the plot legend (b) Top panel: Box plot showing $I_{D,max}$ measured for identical OECTs for insertion of different anions. Bottom panel: Mean $I_{D,max}$ as a function of the ionic radius of the inserted anion. Error bars represent standard deviation. (c) Five cycles of CV at a scan rate of 100 mV/s in electrolytes containing each of the four anions from 0.2 V to $V_{CV,max}$ corresponding to $V_{G,max}$ used for OECT measurements in Fig. 2a. (d) Mean charge under the reductive wave of the CVs and corresponding volumetric doping density. Dashed black lines indicates the charge that would correspond to charging of typical electrostatic double layer ($C_{EDL} = 40 \mu\text{F}/\text{cm}^2$) at respective $V_{CV,max}$

8) The authors write that V_g was swept to values up to which stable OECT operation was observed. What happens at higher gate voltage values?

Beyond the maximum gate voltage values used for each of the ions, we observed that drain currents decrease and does not recover over subsequent cycles. Therefore gate voltages were limited to the stable operating gate voltage range for all measurements performed in this work.

Reviewer 3 remarks

01. The manuscript lacks novelty to be published in Nature Communications. Side chain dependent electrochemical doping of conjugated polymers has been studied extensively over the past years, especially in OECTs, with similar observations and conclusions.

We respectfully disagree with the reviewer regarding the novelty of this paper. We would like to note that this is the first attempt at providing a mechanistic insight for designing polymers with intrinsic abilities to enable ambipolar mixed ionic electronic conduction (MIEC). The importance of our work cannot overstated – for example, there are only 2 known polymers that afford ambipolar MIEC. The paucity of polymers with the ability for ambipolar doping and charge transport is at least in part due to the lack of a reliable mechanistic model and/or the availability of descriptors for designing polymers with the necessary attributes for ambipolar MIEC. Our work provides a framework and also suggests potential side-chain functional groups that could guide the design of high performance polymers with desired MIEC.

While the reviewer is right about reports on side-chain dependence of OECT operation, the consensus of these reports is that OECT materials design requires glycol side-chains to increase hydrophilicity of the polymer matrix that enables passive swelling and facilitates ion insertion. However, passive swelling has a definite negative effect on the carrier transport due to disruption of electronic pathways (Savva, A. *et al. Chem. Mater.* **31**, 927–937 (2019)). The general consensus is that this is an unavoidable trade-off (Savva, A. *et al. Chem. Mater.* **31**, 927–937 (2019)). A few reports showed ion insertion into otherwise hydrophobic polymers but did not detail the mechanism or how it could be generalized to inform OECT materials design rules (Surgailis, J. *et al., Adv. Funct. Mater.* **31**, 2010165 (2021), Nicolini, T. *et al., Adv. Mater.* **33**, 2005723 (2021))

In this report, we suggest that hydrophilicity is not necessary for achieving ambipolar dopability. In fact, we show that 2DPP-OD-TEG with 20% by weight of TEG side chains is effective at coordinating both cations and anions while not showing any passive swelling. Our proposed DN/AN framework will help screen potential side-chain functional groups to

incorporate into future OECT materials. Clearly, our work goes beyond a simple extensive study of side-chain dependent electrochemical doping behaviour. Our views are also echoed by the comments of Reviewer 1 and 2.

02. The proposed hypothesis lacks generality. While the Gutman donor and acceptor numbers provide a qualitative explanation for the doping behavior of the two DPP polymers presented in the study, the hypothesis has not been validated with other functional groups or polymers.

It seems that the reviewer has discounted the validation of our theory with 2DPP-OD-HEX. We would like to remind that the mechanistic theory was developed based on an extensive set of electrochemical experiments with ions of different sizes using 2DPP-OD-TEG as the channel material. We observed an asymmetric charge-polarity dependent ion-insertion in 2DPP-OD-TEG – to the best of our knowledge there is no existing theory that could have predicted or explained our observations.

Therefore, in this work, we proposed that ion-polymer interactions can be modelled akin to ion-solvent interactions. Based on this proposed theory, we could explain the observed charge-polarity dependent asymmetry in 2DPP-OD-TEG. Further, the theory predicted that electrochemical doping in 2DPP-OD-HEX should not have this asymmetry. We validated this prediction from the theory with experiments.

We have also suggested in Fig. 6, some side-chain functional groups that could enable ambipolar electrochemical doping. Conjugated-polymers with these functional groups have not been synthesized before – most commercial conjugated polymers have alkyl side chains. While we intend to build on this work to develop other polymers with ambipolar dopability, the synthesis, characterization and subsequent electrochemical doping of an entirely novel class of polymers is clearly beyond the scope of this work.

03. The manuscript fails to describe how DN and AN of polymers can be quantified, especially for copolymers with different side chains. For example, in the case of the 2DPP-OD-TEG polymer, TEG side chains make up just 20% of the total polymer weight, and the cation coordinating power (DN) is said to be similar to that of water. How much can the TEG : Alkyl side chain ratio be tuned to obtain different ion-dependent doping properties?

In general, the number of side-chains that enable ion coordination should be of the order of the number of dopants. Specifically, in the case of 2DPP-OD-TEG, we would like to note that 20 wt% TEG side chains is a significant number. 20 wt% TEG side chains is equivalent to a number density of $\sim 8 \times 10^{20}$ TEG chains per cm^3 of the polymer (mol. wt. of repeat unit = 1480.19 g/mol and assuming density of polymer = 1 g/cm^3). The calculated volumetric density of inserted cations during n-type doping is in the range of 1.2×10^{20} to 2.1×10^{20} . In other words, there are 4-8 TEG side-chains for every dopant ion and this observation indicates a few TEG side-chains per dopant ion are enough to sufficiently stabilize the dopant ion inside the polymer matrix making ion-insertion energetically feasible.

Therefore, as long as the coordinating side-chain number density is of the order of the dopant density, ion-insertion should be energetically feasible. The exact number of side-chains needed per dopant ion can be calculated based on the coordination environment of the ion. This places a lower bound on the coordinating side-chain fraction. Based on our work and some previous studies, the upper bound on the number density of side-chains should be below the level that favours passive solvent uptake.

Since it is the coordinating side-chains that dictate ion-insertion energetics, the donor number and acceptor number of relevance are the DN/AN of the coordinating side-chains. This is clearly discussed in page 16 of the main text.

The following discussion regarding the relation between side-chain number density and electrochemical dopant density has been added to page 20 of the revised manuscript.

“N-type dopant densities at -1 V calculated from CVs range from 1.2×10^{20} to 2.1×10^{20} for the different cations used in this study. A side-chain number density of the same order is sufficient to interact with and stabilize the dopant ions in the polymer matrix. For example, in 2DPP-OD-TEG used in this study, TEG side-chains make up $\sim 20\%$ of the total weight of the polymer. This translates to $\sim 8 \times 10^{20}$ TEG side-chains per cm^3 which corresponds to 4-8 TEG side-chains per cation (For these calculations, molecular weight of polymer repeat unit (Fig 1a) is taken as 1480.19 g/mol and the density of the polymer is assumed to be $= 1 \text{ g/cm}^3$). Thus, 20 wt% TEG side chains in 2DPP-OD-TEG is a sufficiently large side-chain density to enable dopant-ion stabilization via side-chain coordination. We note that we did not observe any passive swelling of 2DPP-OD-TEG (also see Supplementary Fig. 19).”

04. Polymer dopability is argued in terms of how well the ions can be coordinated by the side chains compared to solvation by the electrolyte. What are the possibilities of solvated ions entering the polymer during electrochemical doping?

Whether the ions are completely desolvated during ion insertion or if they are inserted with a part of the original solvation shell does not change the conclusions of our work. Ions retaining a portion of the solvation shell can be considered to be ‘dressed’ ions with an effectively lower sensitivity to the change in AN/DN values. For the specific case of cations, for example, the insertion of solvated ions is equivalent to having a lower a_{A^+} in the equation:

$$\Delta G_{A^+,sol \rightarrow Poly}^{transfer} = a_{A^+} \cdot (DN_{Poly} - DN_{sol})$$

These conclusions are also consistent with the observed differences for the insertion of Na^+ and Cl^- . If at all, hydration energy is a defining parameter, there should not have been any differences in ion-insertion tendency for Na^+ and Cl^- as they have similar hydration energies. Further, if glycol side-chain induced hydrophilicity of the polymer is the only relevant parameter, we cannot explain the stark difference in the ion insertion of these two similar (in terms of hydration energy) but oppositely charged ions. This charge-polarity dependent asymmetry for Na^+ and Cl^- ion-insertion into 2DPP-OD-TEG is a clear indication of ion-specific ion-side-chain interactions playing an important role in electrochemical doping of π -CPs. The following text was added in page 16 of the revised main manuscript.

“Ions could retain a part of their solvation shell during ion insertion. In this case, the ions can be treated as ‘dressed’ ions with a higher effective radius and a lower effective Lewis acidity/basicity, depending on ionic charge. In case of cations, such co-insertion of solvent molecules can be accounted by readjusting the a_{A^+} parameter, such that a_{A^+} increases with increasing effective radius and decreasing Lewis acidity.”

05. Designing polymers with high DN/AN side chains might not always be beneficial as mentioned in the manuscript. Strong ion attraction can cause ion trapping and limit the reversibility of the doping process.

We agree with the reviewer that at extremely high DN/AN values, a strong ion coordination might limit ionic-conductivities. However, at what values of DN/AN does this occur and how does this vary with ionic size needs to be studied. Conjugated polymers with DN/AN values of

polymeric side chains that are close to the DN/AN values of water could be the best targets for developing high performance OECTs without the need for performance-degrading passive swelling. We have updated the revised main text as follows in page 16.

“We anticipate that π -CPs functionalized with side-chains whose DN and AN are close to that of water could enable aqueous operable ambipolar electrochemical devices including single-component OECT based logic devices⁴⁹, dual-ion batteries⁵⁰ and electrochemically-active elements for neuromorphic computing⁵¹. The exact DN/AN values may be chosen based on the polymer and the dopant ion, keeping in mind that extremely strong ion-side-chain interactions might lead to ion trapping effects.⁵² In general, the DN/AN framework provides a powerful tool to choose appropriate side-chain functional groups for high performance mixed ionic electronic conductors.”

REVIEWERS' COMMENTS

Reviewer #1 (Remarks to the Author):

The manuscript, which I already thought represented a valuable contribution to the field, has been improved in response to reviewer comments. The concerns that I had have been appropriately addressed, as have the concerns of other reviewers. I am happy to recommend this manuscript for publication.

Comments:

The reason for using Pt electrodes is something that largely goes unmentioned in the field. Adding one sentence to the manuscript that states Pt was used instead of gold because it is more stable at high potentials would be useful.

Although I understand the explanation provided, I maintain that working this far outside of the electrochemical stability window of your electrolyte is unusual and recommend the authors restrict their voltage window or change electrolyte for future studies.

Reviewer #2 (Remarks to the Author):

The authors have convincingly addressed the reviewers' comments and provided additional experimental data to support the original claims. For these reasons, I recommend accepting this manuscript in its present form.

Title: Charge Polarity-Dependent Ion-Insertion Asymmetry During Electrochemical Doping of an Ambipolar π -Conjugated Polymer

Authors: Jibin J Samuel, Ashutosh Garudapalli, Chandrasekhar Gangadharappa, Smruti Rekha Mahapatra, Satish Patil, Naga Phani B. Aetukuri*

Response to Reviewer Comments

We thank the reviewers for their feedback and interest in our work. We reproduced the reviewer's comments in blue colored font and our responses to the comments in normal type font. The changes made in the revised manuscript in response to review comments are highlighted in yellow. The comments are listed and addressed point-by-point below:

Reviewer #1 (Remarks to the Author):

The manuscript, which I already thought represented a valuable contribution to the field, has been improved in response to reviewer comments. The concerns that I had have been appropriately addressed, as have the concerns of other reviewers. I am happy to recommend this manuscript for publication.

We thank the reviewer for their positive assessment of our work and their recommendation for publication of this manuscript.

Comments:

The reason for using Pt electrodes is something that largely goes unmentioned in the field. Adding one sentence to the manuscript that states Pt was used instead of gold because it is more stable at high potentials would be useful.

We have updated the main manuscript with the following text in paragraph 3 of page 4 in the main text.

“We found that gold contacts were not electrochemically stable at such high oxidative potentials. Therefore, platinum contacts were used for all OECTs in this work.”

Although I understand the explanation provided, I maintain that working this far outside of the electrochemical stability window of your electrolyte is unusual and recommend the authors restrict their voltage window or change electrolyte for future studies.

We agree with the reviewer that working at voltages far outside the electrochemical stability window of water will have implications for stable OECT operation. However the objective of this work is to understand ion insertion dynamics which do not necessarily require long term device stability. We have made a note of the reviewer's recommendation and will adhere to this in future studies relating to OECT applications.

Reviewer #2 (Remarks to the Author):

The authors have convincingly addressed the reviewers' comments and provided additional experimental data to support the original claims. For these reasons, I recommend accepting this manuscript in its present form.

We thank the reviewer for their positive assessment of our work and their recommendation for publication of this manuscript.